JCB Journal of Cell Biology

# Early proteostasis of caveolins synchronizes trafficking, degradation, and oligomerization to prevent toxic aggregation

Frederic Morales-Paytuví[1], Alba Fajardo[1], Carles Ruiz-Mirapeix[1], James Rae[2], Francesc Tebar[4], Marta Bosch[1,4], Carlos Enrich[4], Brett M. Collins[2], Robert G. Parton[2,3], and Albert Pol[1,4,5]

**Caveolin-1 (CAV1) and CAV3 are membrane-sculpting proteins driving the formation of the plasma membrane (PM) caveolae. Within the PM mosaic environment, caveola assembly is unique as it requires progressive oligomerization of newly synthesized caveolins while trafficking through the biosynthetic-secretory pathway. Here, we have investigated these early events by combining structural, biochemical, and microscopy studies. We uncover striking trafficking differences between caveolins, with CAV1 rapidly exported to the Golgi and PM while CAV3 is initially retained in the endoplasmic reticulum and laterally moves into lipid droplets. The levels of caveolins in the endoplasmic reticulum are controlled by proteasomal degradation, and only monomeric/low oligomeric caveolins are exported into the cis-Golgi with higher-order oligomers assembling beyond this compartment. When any of those early proteostatic mechanisms are compromised, chemically or genetically, caveolins tend to accumulate along the secretory pathway forming non-functional aggregates, causing organelle damage and triggering cellular stress. Accordingly, we propose a model in which disrupted proteostasis of newly synthesized caveolins contributes to pathogenesis.**

## Introduction

Caveolins are membrane-sculpting proteins driving the formation of caveolae, dynamic bulb-shaped invaginations on the plasma membrane (PM) of most vertebrate cells (Parton, 2018). Caveolin-1 (CAV1) in most cell types and CAV3 in cardiac and skeletal muscles are indispensable caveolar components. Caveolae have been implicated in mechanoprotection, signal transduction, lipid homeostasis, and endocytosis (Cheng and Nichols, 2016). Furthermore, by yet poorly understood mechanisms, caveolins function independently of caveolae-regulating lipid-dependent processes in the endoplasmic reticulum (ER), lipid droplets (LDs), mitochondria, and endosomes (Pol et al., 2020).

Within the mosaic environment of the PM, caveola formation is unique as it requires gradual oligomerization of hundreds of newly synthesized caveolin protomers during their biosynthetic transport from the ER, through the Golgi complex (GC), and to the PM (Han et al., 2016b). Early studies suggested that caveolin protomers are co-translationally inserted into the ER through the central hydrophobic domain in a signal-recognition particle (SRP)-dependent manner to adopt a hairpin topology with both ends facing the cytosol (Monier et al., 1995). Studies such as

these, employing in vitro import into microsomes, suggested that caveolins spontaneously oligomerize into 8S complexes (~11–14 protomers) that are stabilized by free cholesterol or fatty acids (Han et al., 2016b; Monier et al., 1995, 1996). Current schemes suggest that the 8S complexes would be rapidly concentrated in ER-exit sites to be transported by COPII vesicles to the GC and assemble 70S oligomers (~150 protomers) that could reorganize the bilayer to accrue an estimated 20,000 cholesterol molecules (Hayer et al., 2010a; Ortegren et al., 2004). Once formed, the 70S oligomers rapidly traffic as caveolar carriers from the trans-GC network to the PM to form long-lived but metastable caveolae (Kovtun et al., 2015; Tagawa et al., 2005). Furthermore, other high-ordered caveolin oligomers have been observed on the PM (Khater et al., 2019; Lolo et al., 2023), suggesting alternative oligomerization and trafficking mechanisms functioning in the GC.

Reflecting the complexity of the biosynthetic steps followed by caveolins, the most naturally occurring caveolin mutations that underlie disease appear to lead to the mis-assembly of caveolin complexes during these early trafficking steps. Most

[1]Lipid Trafficking and Disease Group, Institut d'Investigacions Biomèdiques August Pi i Sunyer (IDIBAPS), Barcelona, Spain; [2]Institute for Molecular Bioscience (IMB), The University of Queensland (UQ), Brisbane, Australia; [3]Centre for Microscopy and Microanalysis (CMM), The University of Queensland (UQ), Brisbane, Australia; [4]Department of Biomedical Sciences, Faculty of Medicine, Universitat de Barcelona, Barcelona, Spain; [5]Institució Catalana de Recerca i Estudis Avançats (ICREA), Barcelona, Spain.

Correspondence to Robert G. Parton: r.parton@imb.uq.edu.au; Albert Pol: apols@ub.edu.

caveolinopathies are defined as ER-GC disorders characterized by the intracellular accumulation of caveolins and the formation of non-functional caveolin aggregates interfering with a variety of intracellular processes such as ER-GC homeostasis, mitochondrial function, cytoskeleton organization, and extracellular matrix formation (Gonzalez Coraspe et al., 2018). These defects lead to disease conditions such as pulmonary arterial hypertension and congenital generalized lipodystrophy in the case of CAV1 mutants or sarcolemma damage and muscular dystrophies for CAV3 mutants (Copeland et al., 2017; Gonzalez Coraspe et al., 2018).

It is unknown if progressive disruption of the early proteostasis of wild type (wt) caveolins, for example in the environment of unhealthy cells, is deleterious through analogous mechanisms to those triggered by caveolin mutants. If this was the case, it could suggest a unifying theme explaining the unresolved contribution of wt caveolins to a variety of long-term diseases characterized by generalized failures in proteostasis like senescence (Volonte and Galbiati, 2020), diabetes (Haddad et al., 2020), and neurodegeneration (Yang et al., 2020). Understanding how dysfunctional caveolin biosynthetic trafficking could lead to pathogenic conditions initially requires a holistic characterization of their poorly understood early proteostasis.

Here, we have optimized experimental protocols for comparing the initial biosynthetic steps of the caveolin family members and paradigmatic pathogenic mutants. We have observed a previously unappreciated complexity in the regulation of newly synthesized caveolins. These proteostatic mechanisms include signals to modulate ER retention or export, to balance anterograde trafficking with proteasomal degradation, and to avoid oligomerization until caveolins reach the environment of the medial- or trans-GC. When proteostasis fails, due to chemical inhibitors or specific mutations, instead of assembling into functional oligomers, wt and mutant caveolins accumulate intracellularly, tend to form non-functional aggregates, cause organelle damage, and trigger cellular stress. Accordingly, we propose a model in which the disrupted early proteostasis of caveolins contributes to pathogenesis.

## Results

### In silico modeling of caveolin oligomers

While many previous studies have suggested that caveolins possess distinct functional domains (Fig. 1, A and B), the CAV1 structure recently determined by cryoEM provides new insights into their assembly (Porta et al., 2022). Recombinant CAV1 forms a ring-shaped oligomer composed of 11 primarily α-helical subunits that spiral toward a central β-barrel formed by their C-terminal sequence (8S oligomer). This unique architecture generates a flat disc with one extremely hydrophobic face that likely penetrates the cytosolic leaflet of the lipid bilayer (Parton and Collins, 2022; Porta et al., 2022).

To make additional predictions about the structure of the caveolin oligomers, we generated in silico models with AlphaFold2 (Evans et al., 2022 Preprint; Jumper et al., 2021). Using CAV1 as a reference (Fig. 1 B), the algorithm predicted that caveolins form essentially identical inter-subunit structures to those observed by cryoEM (Fig. 1 C). The expected oligomer also involves 11 primarily α-helical subunits that organize forming a flat disc penetrating the cytosolic leaflet of the lipid bilayer (Fig. 1 D). AlphaFold2 anticipates that CAV2 and CAV3 organize into similar oligomeric structures (Fig. 1, E and F; and Fig. S1).

The disc-shaped topology of tightly packed protomers provides a potential explanation for the observation that CAV1 on the PM is only recognized by antibodies raised against N-terminal residues, an accessible region that is not part of the ring but extends into the cytoplasm (Hayer et al., 2010a; Luetterforst et al., 1999; Pol et al., 2005). In contrast, antibodies raised against the central signature motif or against the C-terminus do not recognize CAV1 on the PM but only a pool of CAV1 within the cis-GC (Hayer et al., 2010a; Luetterforst et al., 1999; Pol et al., 2005). Hence, in contradiction to previous schemes (Han et al., 2016b), these observations, the cryoEM analysis, and AlphaFold2 anticipate that the central caveolin signature motif remains exposed in the early GC, whereas the fully-assembled disc-shaped oligomers only form beyond this compartment during anterograde transport.

In support of the idea that oligomers assemble once caveolins are in the GC, the ER exit motifs previously found to act as COPII binding sequences in CAV1 ($_{67}$DFE$_{69}$) and CAV3 ($_{40}$DFE$_{42}$; Hayer et al., 2010a) are also expected to be physically inaccessible when assembled in a disc (Fig. 1 F). These three residues are tightly packed as part of the inter-subunit interactions required for oligomerization. The central Phe side chain is stacked within a hydrophobic pocket while the flanking Asp and Glu side chains are shielded by the Pin motif of the adjacent protomer. Hence, if caveolin oligomers were formed in the ER are unlikely to be transported to the GC, this would require in vivo mechanisms to avoid spontaneous oligomerization of caveolins in the ER described in vitro using purified microsomes (Monier et al., 1995, 1996).

### Differential biosynthetic trafficking of caveolin family members

To test these predictions, we adapted methods previously used for characterizing the early biosynthesis of caveolins (Hayer et al., 2010a). Unlike these studies, we N-terminally tagged caveolins to avoid interfering with the recently demonstrated key role of the C-termini in the formation of the 8S oligomers (Porta et al., 2022; Fig. 1 C). In contrast, the N-terminus of caveolins extends into the cytosol and is not directly involved in the assembly of the ring (Fig. 1, E and F; and Fig. S1). Thus, GFP-CAV1, CAV2, or CAV3 were transfected into COS-1 cells in the presence of cycloheximide to inhibit protein synthesis. After 6 h, the drug was removed to allow synchronized protein expression for 3 h (Fig. 2 A). We selected 3 h to avoid potential artifacts only observed with longer expression times (Hayer et al., 2010a).

First, protein expression was confirmed by immunoblotting with anti-GFP antibodies (Fig. 2 B). Protein levels were significantly different, with CAV1 consistently expressed at the highest level and CAV3 at the lowest (Fig. 2 C). Next, the intracellular distribution of caveolins was analyzed by fluorescence microscopy. As expected for proteins following the biosynthetic-secretory pathway, the newly synthesized caveolins are

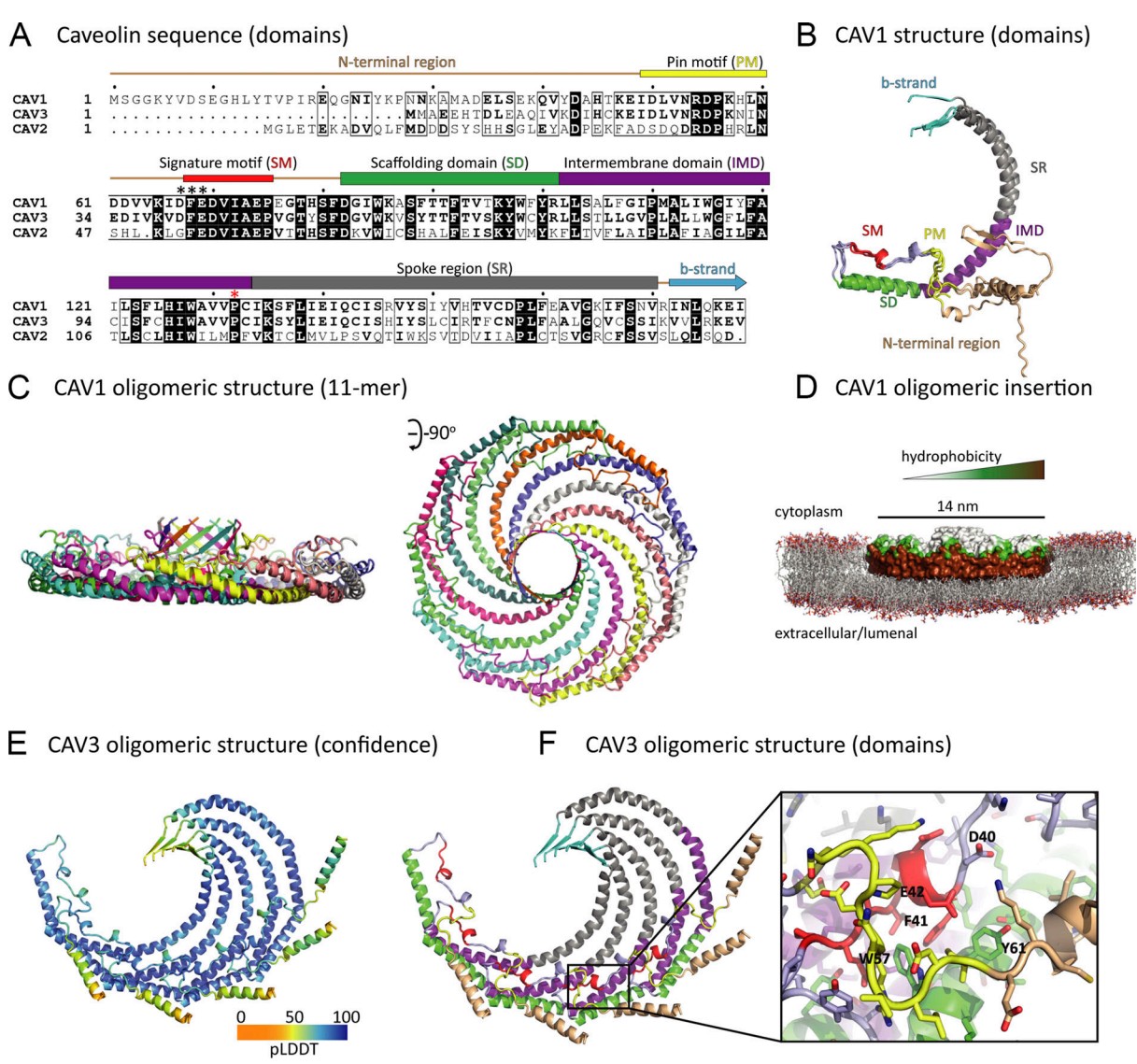

Figure 1. **Structure of CAV1 and modeling of CAV2 and CAV3. (A)** Sequence alignment of human CAV1, CAV2, and CAV3. Regions of the structure are indicated above and colored: SM, signature motif (red); SD, scaffolding domain (green); and IMD, intermembrane domain (purple); PM, pin motif (yellow); SR, spoke region (gray); and beta-strand (cyan). The "DFE" sequence of CAV1 and CAV3 mediating COPII binding and ER export is highlighted by asterisks (*). The site of pathogenic Pro to Leu mutation (P132L in CAV1 and P104L in CAV3) is indicated by a red asterisk. **(B)** Structural alignment of an individual monomer of CAV1 from AlphaFold2 predictions. The structure is colored using the same scheme as Porta et al. (2022) with the positions of regions labeled as in A. The previously proposed oligomerization domain (OD) contains the SM and SD. **(C)** Structure of the CAV1 11 subunit oligomer determined by cryoEM (Porta et al., 2022; see AlphaFold prediction in Fig. S1). Each CAV1 protomer is shown in cartoon representation in a different color. Note that only residues from 49–178 were resolved in the cryoEM maps. **(D)** CAV1 cryoEM structure shown in surface representation colored according to hydrophobicity. The CAV1 coordinates were docked into a model lipid bilayer using PyMOL to show the scale of the protein complex compared to the membrane, and to visualize the proposed mechanism of membrane docking and insertion into the cytoplasmic leaflet. **(E and F)** Five subunits of full-length CAV3 modeled with AlphaFold2. In E, protomers are colored according to the pLDDT scores, and in F, according to the domains depicted in A and B. The CAV3 (40DFE42) inter-subunit interactions are detailed. The central Phe (F) side chain is stacked within a hydrophobic pocket while the flanking Asp (D) and Glu (E) side chains are shielded by the PM motif of the adjacent protomer.

distributed between different compartments. CAV1 was visible in the GC and PM of most cells (Fig. 2 D and Fig. S2). In contrast, CAV2 mainly accumulated in the GC (Fig. 2 E and Fig. S2). At this timepoint, CAV3 was commonly observed in the GC but, in contrast to CAV1, mainly distributed between the ER and LDs with rare PM labeling (Fig. 2, F and G; and Fig. S2). The distribution of newly synthesized GFP-tagged caveolins was quantified in confocal microscopy images using specific antibodies for these organelles. The Manders overlap coefficient (M2) confirmed the differential early trafficking. Although the three caveolins colocalized with TGN46 (GC marker), CAV1 demonstrated a strong overlap with Cavin-1 (PM caveolae marker) while CAV3 significantly overlapped with Calreticulin (ER marker) and Perilipin-3 (Plin3, LD marker; Fig. 2 H). The differential trafficking of CAV1 and CAV3 was also observed with a small N-terminal Myc tag (results not shown) and was even

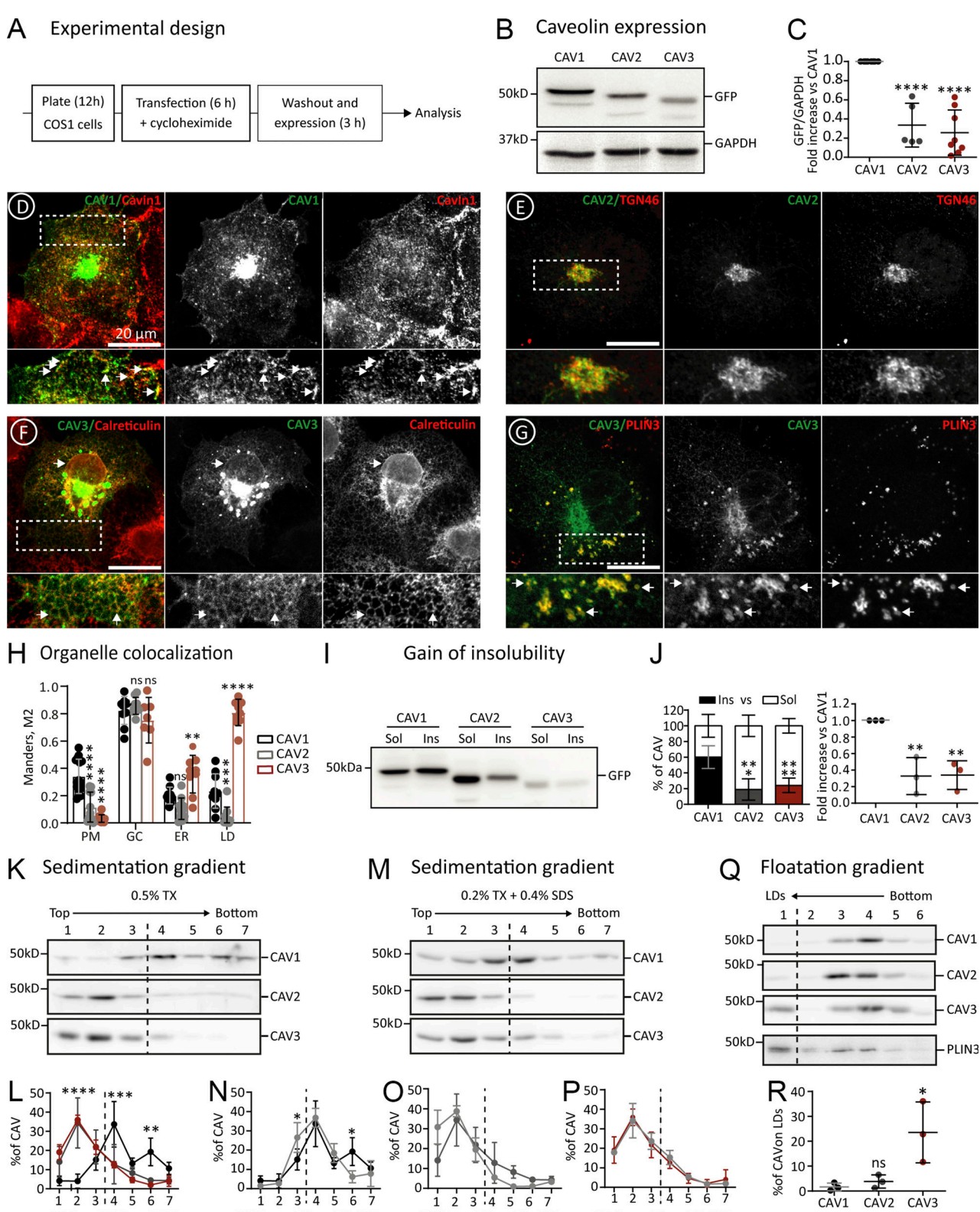

Figure 2. **Differential proteostasis of newly synthesized caveolins. (A)** Experimental design for studying the early proteostasis of caveolins. This protocol has been used throughout the work for the transfection (1 µg of DNA), expression (for 3 h), and analysis of caveolins and mutants, unless otherwise specified. **(B and C)** N-terminally GFP-tagged CAV1, CAV2, or CAV3 were transfected in parallel and after 3 h protein steady-state levels were determined in cell lysates (equal protein concentration) by immunoblotting (IB) with anti-GFP antibodies (B). Band intensity was quantified by densitometry, corrected with respect to the GAPDH levels determined by IB in each sample, and expressed relative to the GFP levels in CAV1 transfected cells (C; n ≥ 5 independent experiments). **(D–H)** The intracellular distribution of GFP-caveolins after 3 h was analyzed by confocal microscopy using specific antibodies against organellar markers; Cavin-1 (PM caveolae), TGN46 (GC), Calreticulin (ER), and Plin-3 (LDs). Representative images are included (D–G and Fig. S2). GFP-caveolin colocalization with each

organelle was quantified using the Manders M2 overlapping coefficient (*n* ≥ 3 independent experiments with at least seven cells per condition; H). **(I and J)** Cells transfected with the indicated proteins were homogenized after 3 h with 1% Triton X-100 (4°C) and soluble and insoluble fractions separated by centrifugation. Protein distribution was analyzed in equal volumes of each fraction by IB with anti-GFP antibodies. **(I and J)** A representative result (I) and the relative distribution (J) of the protein in both fractions (left panel), and the relative insoluble fraction of CAV2 and CAV3 when compared with CAV1 (right panel; *n* = 3 independent experiments). **(K–P)** Transfected cells were solubilized after 3 h with either 0.5% Triton X-100 (TX) or 0.2% TX-100 + 0.4% SDS (TX+SDS), loaded at the top of sucrose density gradients, and fractionated by centrifugation according to their molecular weight. Protein distribution was quantified in equal volumes of each fraction by IB with anti-GFP antibodies. **(K and M)** show a representative result and **(L–P)** the relative distribution of the proteins in each fraction of the gradients quantified by densitometry (*n* ≥ 3 independent experiments). **(Q and R)** Transfected cells were homogenized after 3 h by nitrogen cavitation in the absence of detergents, loaded at the bottom of sucrose gradients, and fractionated according to their buoyant density by centrifugation for 1 h. Protein distribution was calculated in equal volumes of each fraction by IB with anti-GFP antibodies (*n* = 3 independent experiments). Upon these conditions, LDs float into the top fraction, as corroborated by enrichment in endogenous Plin-3. **(Q and R)** A representative result (Q) and the relative enrichment (R) of each caveolin on LDs. All graphs show means ± SD; ns, not significant; *P < 0.05, **P < 0.01, ***P < 0.001, ****P < 0.0001 calculated in a one-way ANOVA (C, J, and R) or two-way ANOVA tests (H, L, N, O, P, and Q). Scale bars are 20 µm. Source data are available for this figure: SourceData F2.

more striking in a model muscle cell line (C2C12 myoblasts), suggesting that this differential trafficking is not dependent on the cell type (Fig. S2). Thus, although CAV1 and CAV3 are largely indistinguishable and accumulate on the PM when expressed for 24 h (Luetterforst et al., 1999; Pol et al., 2001), our analysis at very early time points suggests that this trafficking is significantly slower for CAV3.

The biosynthetic transport of caveolins from the sites of synthesis to the PM is associated with gradual changes in their biochemical traits. Hence, the observed differential proteostasis of newly synthesized caveolins was further validated and quantified by applying biochemical criteria. Transport of caveolins to PM caveolae is associated with the acquisition of insolubility to 1% Triton X-100 (1% TX at 4°C; Simons and Ikonen, 1997). To corroborate the distinct arrival of caveolins to the PM, transfected cells were solubilized in cold 1% TX and fractionated into insoluble pellets and soluble supernatants (Pol et al., 2001). In contrast to CAV2 and CAV3, only CAV1 becomes significantly insoluble when expressed for 3 h, confirming that CAV1 is efficiently transported to the PM (Fig. 2, I and J; and Fig. S2 J).

The gain of insolubility of caveolins is mediated by their gradual oligomerization. To determine the oligomeric state, transfected cells were solubilized with mild 0.5% TX, loaded at the top of density gradients, and oligomers were sedimented for 16 h to reach the equilibrium according to their molecular weight (Copeland et al., 2017; Hayer et al., 2010a). Only CAV1 sedimented into high-density fractions, likely reflecting its more efficient oligomerization (Fig. 2, K and L). Transfected CAV1 tended to accumulate around two peaks (fractions #4 and #6), previously recognized as the 8S and 70S oligomers. In contrast, CAV2 and CAV3 remained at the top of the gradients (fractions #1 to #3), reflecting their monomeric/low oligomeric state and confirming that not all caveolins spontaneously oligomerize after expression in the ER.

To confirm that GFP-CAV1 assembles into oligomers rather than into non-functional high-molecular-weight aggregates, like those formed by mutants failing to oligomerize (details in Figs. 5 and 6), cells were solubilized with a combination of TX and sodium dodecyl sulfate (0.2% TX and 0.4% SDS; TX+SDS). Upon these conditions, the 8S oligomers remain as insoluble complexes but the 70S oligomers and the aggregates are largely solubilized into 8S oligomers or monomeric caveolins, respectively (Copeland et al., 2017). When transfected cells were homogenized in TX+SDS and sedimented in gradients, CAV1

distributed around a single high-molecular-weight peak corresponding to the 8S structure (fraction #4), indicating that CAV1 assembles into functional oligomers (Fig. 2, M and N). In contrast, as expected for monomeric or low-oligomeric caveolins, CAV2 and CAV3 sedimented identically in both types of gradients (Fig. 2, M, O, and P).

Finally, to independently assess the distribution of newly synthesized caveolins with respect to LDs, transfected cells were homogenized by nitrogen cavitation in the absence of detergents, loaded at the bottom of sucrose gradients, and floated for just 3 h (velocity gradients; Bosch et al., 2020b; Turro et al., 2006). Only CAV3 floated to the top LD-containing fraction, reflecting its high affinity for LDs (Fig. 2, Q and R).

Thus, at early timepoints after synthesis, CAV1 rapidly exits the ER, oligomerizes in the GC, gains insolubility, and traffics to the PM. In contrast, CAV2 is efficiently exported from the ER but mostly accumulates in the GC without forming oligomers whereas CAV3 is retained for longer periods in the ER, remains in a low oligomeric state without spontaneously assembling into 8S oligomers, and laterally traffics to LDs.

## Proteasomal degradation of newly synthesized caveolins avoids aggregation and trafficking defects

The lower CAV3 levels when compared with CAV1 could be the result of greater exposure to the degradation mechanisms functioning in the ER. Active degradation could explain why caveolins do not spontaneously oligomerize/aggregate following synthesis in vivo but they do in vitro. We focused on CAV1 and CAV3 since CAV2 is unable to drive caveola formation alone (Parolini et al., 1999).

To test if caveolins are degraded early after synthesis, CAV1 and CAV3 were expressed for 3 h in the presence of the proteasomal inhibitor MG132. Inhibition led to increased caveolin levels (Fig. 3, A and B). GFP-ALDI (METTL7B, an ER/LD resident protein) was also sensitive to MG132, but GFP-VAMP-associated protein A (VAP-A, transmembrane ER/GC) and GFP-Syntaxin 6 (GC soluble protein) were largely unaffected at these time points. Thus, in contrast to PM caveolins, which are long-lived proteins with an estimated half-life from 10 to 36 h that are degraded in lysosomes (Conrad et al., 1995; Hayer et al., 2010b; Ritz et al., 2011), newly synthesized caveolins have a much shorter lifetime and are degraded by the proteasome.

To explore the physiological relevance of the active degradation of newly synthesized caveolins, but avoid the use of drugs

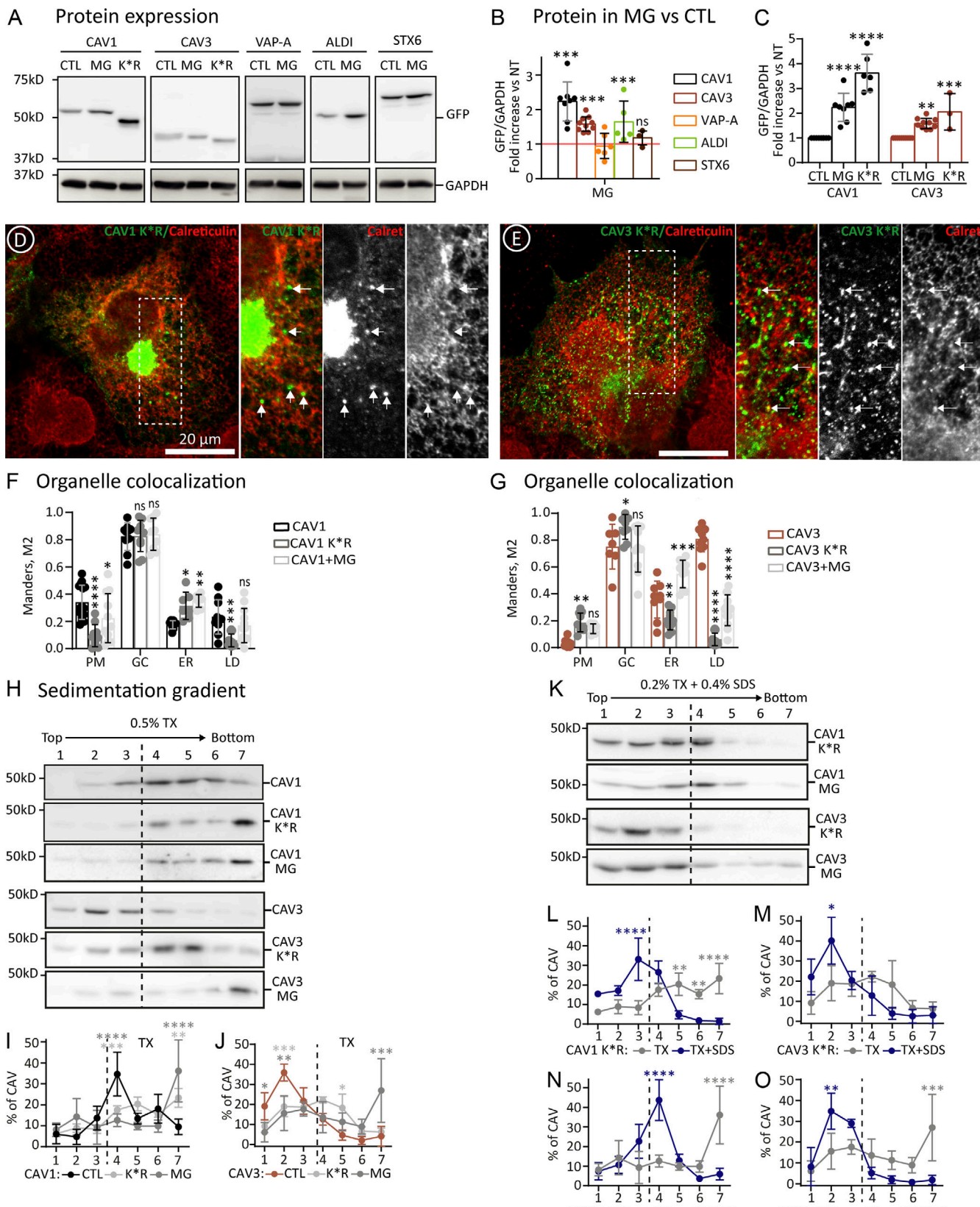

**Figure 3. Proteasomal degradation of newly synthesized caveolins. (A–C)** The indicated proteins were expressed (as in Fig. 2 A) in control (CTL) or in cells additionally treated with MG132 (MG). Protein steady-state levels determined after 3 h by IB with anti-GFP antibodies (A; as in Fig. 2 B). Proteins levels were referred to the expression of each protein in the absence of MG132 (red line; B; n ≥ 4 independent experiments). CAV1 K*R and CAV3 K*R steady-state levels were compared to CTL and MG132-treated cells and referred to the *wt* caveolin (C; n ≥ 3 independent experiments). **(D–G)** CAV1 K*R and CAV3 K*R were

expressed for 3 h, and their intracellular distribution was analyzed by confocal microscopy. D and E show representative images of the mutants and their co-localization with Calreticulin (ER marker). Arrows indicate punctate structures formed by the mutants and are especially apparent along the ER. Additional images are included in Fig. S3. F and G show the colocalization of the mutants with markers of each compartment in confocal microscopy images quantified using the Manders M2 overlapping coefficient (n ≥ 3 independent experiments and at least nine cells per condition). **(H–O)** The indicated proteins were expressed in CTL cells or, in the case of CAV1 and CAV3, in cells additionally treated with MG. After 3 h, cells were solubilized with either TX or TX+SDS and fractionated in sucrose sedimentation gradients (as in Fig. 2, K and M). Protein distribution was determined in equal volumes of each fraction by IB with anti-GFP antibodies. H and K show a representative result and (I–O) the relative distribution of the proteins in each fraction of the gradients (n = 3 independent experiments). All graphs show means ± SD; ns, not significant; *P < 0.05, **P < 0.01, ***P < 0.001, ****P < 0.0001 calculated in a one-way ANOVA test (B and C) and two-way ANOVA test (F, G, I, J, L, and O). Scale bars are 20 μm. Source data are available for this figure: SourceData F3.

affecting other proteins, we analyzed degradation-insensitive caveolin K*R mutants (Lys substitution for Arg impairing ubiquitination). When expressed in cells, these mutants retain the capacity to traffic to the PM (Aslanyan et al., 2023; Hayer et al., 2010b). As expected, after expression for 3 h, CAV1 K*R and CAV3 K*R reached a higher steady-state level than the *wt* proteins (Fig. 3, A and C). However, when compared with CAV1 by microscopy, CAV1 K*R was observed in the ER of most cells, apparently accumulated to a greater extent in the GC (Fig. 3 D), and demonstrated a reduced labeling of the PM, likely reflecting trafficking defects and that the transport of the mutant to the PM is slower than the transport of CAV1 (Fig. 3, D and F; and Fig. S3). In contrast, CAV3 K*R demonstrated a slightly but significantly higher labeling in the PM, likely reflecting the impaired degradation and that more protein pool is available to be transported to the GC. Interestingly, although associated with the ER, CAV3 K*R was largely excluded from LDs, suggesting defective lateral trafficking (Fig. 3, E and G; and Fig. S3).

Trafficking defects of non-degradable caveolins could be caused, as observed in the case of mutants (details in Figs. 5 and 6), by reduced oligomerization and/or protein aggregation. When sedimented in TX gradients, CAV1 K*R equilibrated into very high-molecular-weight species close to the bottom of the gradients (fraction #7; Fig. 3, H and I). Similarly, CAV3 K*R also sedimented into higher-molecular-weight species than CAV3, suggesting that these mutants tend to aggregate (Fig. 3, H and J). Indeed, the high-molecular-weight species formed by the K*R mutants were efficiently solubilized by TX+SDS to render protomers now sedimented at the top of the gradients (Fig. 3, K–M). Morphologically, caveolin aggregates likely correspond to the punctate structures, especially apparent along the ER of cells expressing the K*R mutants but rarely observed in cells expressing CAV1 or CAV3 (Fig. 3, D and E, arrows). When expressed in cells treated with MG132, similarly to the K*R mutants, CAV1 was retained in the ER and displayed reduced association with the PM while CAV3 K*R, although still present in the ER, demonstrated a reduced affinity for LDs (Fig. 3, D–G). In cells treated with MG132, both mutants formed TX-insoluble but TX+SDS-soluble complexes (Fig. 3, H–O), confirming that in the absence of active degradation, caveolins tend to form protein aggregates that lead to trafficking defects.

Thus, newly synthesized caveolins are actively degraded by the proteasome. Active proteasomal degradation avoids the formation of caveolin TX-insoluble but TX+SDS-soluble complexes. When degradation is genetically or chemically inhibited, caveolins display trafficking defects, suggesting that aggregates reduce the pool of caveolins following the biosynthetic-secretory

pathway. That aggregation leads to trafficking defects is clearly illustrated by the common dominant negative capacity of mutated caveolins that, prone to aggregate, intracellularly sequester *wt* caveolins and reduce caveola formation (Copeland et al., 2017; Galbiati et al., 1999; Lee et al., 2002; Pol et al., 2001; Roy et al., 1999).

### CAV3, but not CAV1, contains an ER-retention sequence that determines caveolin steady-state levels

Early proteostasis of caveolins initially involves regulation of their ER retention/export that likely determines its early degradation. Only a minor pool of *wt* CAV1 is visible in the ER, likely suggesting a faster ER export when compared with CAV3 and specific mechanisms to retain CAV3 in the ER. Beyond the expression pattern, this is probably the first time that such marked functional differences between CAV1 and CAV3 have been observed. Thus, we further analyzed these mechanisms and their involvement in the differential proteostasis of each caveolin.

To identify the minimal sequences involved in this differential trafficking, we first confirmed that alpha- and beta-CAV1 (lacking the first N-terminal 32 residues, Fig. 1 A) are identically exported from the ER (Hayer et al., 2010a). Indeed, after 3 h, alpha- and beta-CAV1 were indistinguishable in their distribution between the GC and the PM but were undetectable in the ER (Fig. 4 B). Fractionation of cells solubilized with 1% TX confirmed that both caveolins rapidly gain insolubility and efficiently traffic to the PM (Fig. 4 C). Beta-CAV1 was more stable than alpha-CAV1 (Fig. 4, A and C). Degradation of CAV1 in lysosomes is mediated by ubiquitination on Lys 5, 26, 30, 39, 47, and 57 (Kirchner et al., 2013); as beta-CAV1 lacks several of these Lys (Fig. 1 A), higher protein steady-state levels likely reflect less active ubiquitination and proteasomal degradation in the ER.

Comparison around the conserved COPII binding motif of beta-CAV1 ($_{36}$DFE$_{38}$) and CAV3 ($_{40}$DFE$_{42}$) showed that while the signature and scaffolding domains are practically identical, there is considerable divergence in the N-terminal residues (red residues, Fig. 4 D). This region is enriched in hydrophobic amino acids only in the case of CAV3 and has the propensity to form an alpha-helix according to HeliQuest (Gautier et al., 2008) and AlphaFold2 (Fig. 4 D; Fig. 1, E and F; and Fig. S1). Thus, we determined if these residues retain CAV3 in the ER by exchanging the initial 14 amino acids of beta-CAV1 by the first 16 residues of CAV3 to generate CAV1 with the N-terminus of CAV3 (CAV1 N-CAV3) and CAV3 N-CAV1.

When expressed for 3 h and compared with beta-CAV1 by fluorescence microscopy, CAV1 N-CAV3 showed reduced association with the PM and was detectable in the ER of most cells,

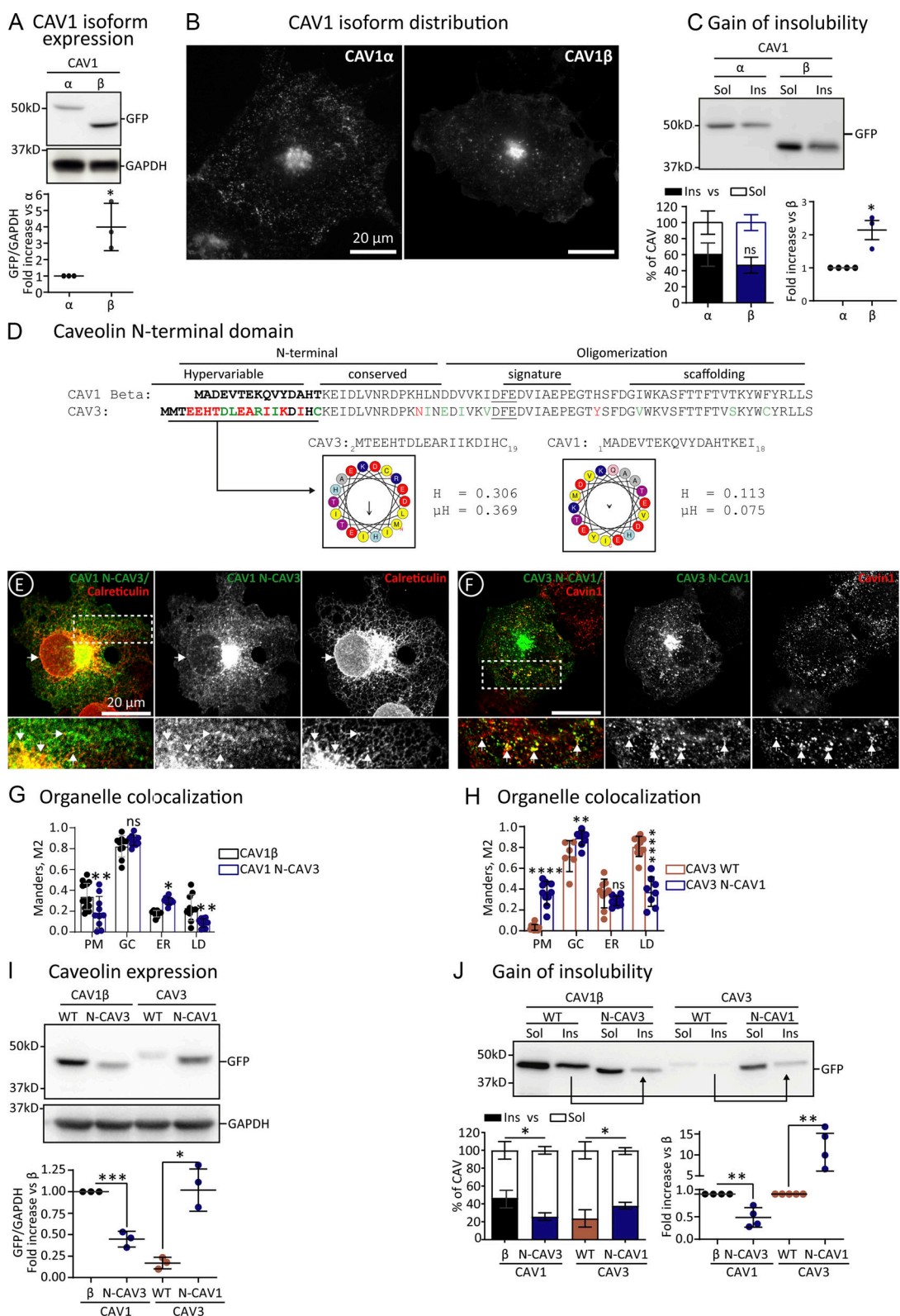

Figure 4. **Molecular determinants for the retention of CAV3 in the ER. (A–C)** GFP-tagged alpha-CAV1 and beta-CAV1 (lacking the initial 32 amino acids of alpha) were expressed for 3 h, and steady-state protein levels were determined by IB with anti-GFP antibodies as in Fig. 2 B (A; *n* = 3 independent experiments). Protein distribution was assessed by confocal microscopy as in Fig. 2 H (B), and the gain of insolubility determined as in Fig. 2, I and J (C). **(D)** The N-terminal domains of beta-CAV1 and CAV3 are aligned, relevant differences in amino acids are indicated in red, and equivalent changes in green. The DFE motif binding COPII is underlined. The hydrophobicity (H), the hydrophobic moment (μH), and the tendency to fold as an alpha helix of the hypervariable domains of both proteins are depicted (hydrophobic residues in yellow and charged amino acids in red and blue). **(E–H)** The initial 14 amino acids of beta-CAV1 were exchanged by the first 16 residues of CAV3 to generate CAV1 with the N-terminus of CAV3 (CAV1 N-CAV3) and CAV3 N-CAV1. The mutants were expressed for 3 h, and

their distribution was analyzed by confocal microscopy as in Fig. 2 D. **(E and F)** show representative images of the mutants and their co-localization with Calreticulin (ER marker) or Cavin-1 (PM marker). Additional images are included in Fig. S4. **(G and H)** show the colocalization of the mutants with organellar markers as in Fig. 2 H and the quantification using the Manders M2 overlapping coefficient (n ≥ 3 independent experiments and at least eight cells per condition). **(I and J)** The indicated proteins were transfected for 3 h and the steady state levels of wt and mutants determined by IB with anti GFP antibodies as in Fig. 2 B. The insolubility to 1% TX of the mutants was determined as in Fig. 2, I and J. Arrows in J indicate the bands showing reduction or gain of insolubility of the mutants. The graph below shows the quantification of the protein levels with GFP antibodies and densitometry with respect to the wt proteins calculated as in Fig. 2, I and J (n ≥ 3 independent experiments). All graphs show means ± SD; ns, not significant; *P < 0.05, **P < 0.01, ***P < 0.001, ****P < 0.0001 in a two-tailed unpaired t test (A and C), one-way ANOVA (I and J), or two-way ANOVA test (G and H). Scale bars are 20 μm. Source data are available for this figure: SourceData F4.

although it demonstrated a low affinity for LDs (Fig. 4, E and G; and Fig. S4). In contrast to CAV3, CAV3 N-CAV1 demonstrated a decreased accumulation of LDs, and it was observable in the PM of most cells (Fig. 4, F and H; and Fig. S4). In agreement with this redistribution, CAV1 N-CAV3 showed reduced steady-state levels, reflecting ER retention and active degradation (Fig. 4 I), and displayed reduced gain of insolubility, indicating reduced oligomerization and arrival to the PM (Fig. 4 J). In contrast, CAV3 N-CAV1 showed a higher steady-state level and insolubility, reflecting reduced ER retention and rapid trafficking to the PM (Fig. 4, I and J).

Hence, when compared with CAV1, an N-terminal alpha-helix retains CAV3 in the ER. Caveolin retention in the ER determines the extent of degradation and the pool of caveolins exported to the GC to traffic to the PM. Intriguingly, CAV1 N-CAV3 does not traffic to LDs, suggesting that in addition to the retention in the ER, other motifs mediate this lateral sorting (Ostermeyer et al., 2001). These mutants confirm that ER retention reduces caveolin steady-state levels and determines the extent of protein available for the biosynthetic-secretory pathway.

**Impaired proteostasis of newly synthesized caveolins causes cellular stress**

Next, we examined the cellular consequences of disrupting the early proteostatic mechanisms functioning on caveolins by applying our experimental systems to the study of mutants affecting different aspects of these mechanisms.

We propose that oligomerization of caveolins is likely occurring in late-GC. To further test this model, we analyzed CAV1 P158PfsX22, a frameshift mutation associated with pulmonary arterial hypertension (Copeland et al., 2017). Although the resulting protein contains intact COPII binding and oligomerization domains, the C-terminus of CAV1-P158 includes a newly generated or unnatural ER-retrieval signal, and thus, the protein is largely excluded from the late-GC (Fig. 5 A). When expressed for 3 h, newly synthesized CAV1-P158, like wt CAV1, was actively degraded by the proteasome, as determined by immunoblotting of transfected cells treated with MG132 (Fig. 5 B). When analyzed by microscopy, in contrast to CAV1, CAV1-P158 accumulated in the ER, LDs, and GC with rare labeling in the PM (Fig. 5, C and D; and Fig. S5). In TX-gradients, CAV1-P158 sedimented as low oligomeric species but also into high-molecular-weight species (Fig. 5 E and F). When expressed in MG132-treated cells, CAV1-P158 demonstrated a remarkable tendency to fractionate at the bottom of the gradients (Fig. 5, E and F), suggesting aggregate formation. Indeed, these high-molecular-weight species were largely solubilized into low oligomeric forms when cells were homogenized with TX+SDS (Fig. 5, E and G). Deletion or

mutation (K to A) of the KKYK motif of CAV1-P158 enables the protein to traffic to caveolae (Copeland et al., 2017), clearly suggesting that retention of caveolins between the ER and the early GC prevents its oligomerization. To determine if the intracellular retention of CAV1-P158 is deleterious for cells, we measured the activation of the unfolded protein response (UPR), an indicator of cellular stress. When compared with wt CAV1, a significant UPR activation was already detected in cells expressing CAV1-P158 as early as 16 h, as quantified by expression of the X-box binding protein 1 (XBP1) and the DNA damage-inducible transcript 3 (DDIT3 or CHOP; Fig. 5 H).

Next, we analyzed caveolin mutants that fail to oligomerize and accumulate in the GC. CAV1-P132L and CAV3-P104L are well-characterized mutants in which a strictly conserved proline residue is substituted by leucine in the hydrophobic domain (Hayashi et al., 2001; Minetti et al., 1998). The pathogenic capacity of these mutants has been commonly attributed to the disruption of the assumed hairpin topology of the caveolin protomer. Theoretically, the substitution of a Pro, inducing a tight turn, by a Leu, forming a straight chain, would disrupt the folding needed to place both caveolin termini facing the cytosol. However, the new structural understanding of the oligomers demonstrates that rather than a hairpin, this conserved Pro plays two other important roles. Using CAV3 as an example (Fig. 6 A), the conserved Pro104 sits at the junction between the intramembrane domain and spoke region and is essential for promoting a kink in the alpha-helical structure between these two elements determining the overall caveolin curvature within the disc. Secondly, Pro104 (from chain i) is packed against Trp87 and Phe91 of the preceding protomer in the ring (chain i-1). Thus, replacing this Pro with the bulkier beta-branched side chain of Leu likely perturbs both curvature and inter-subunit packing, as recently proposed for CAV1-P132L (Han et al., 2023).

The early proteostasis of CAV1-P132L and CAV3-P104L was analyzed after 3 h. Similar to wt caveolins, both mutants were actively degraded by the proteasome, as demonstrated by immunoblotting of transfected cells treated with MG132 (Fig. 6, B and C). Like CAV1, CAV1-P132L was rarely visible in the ER and accumulated in the GC, again consistent with oligomerization being dispensable for ER to GC transport of caveolins (Fig. 6, D and E; and Fig. S5). In contrast to CAV1, CAV1-P132L was not observed on the PM and, accordingly, oligomerization is mandatory for the GC to PM transport of caveolins. Like CAV3, CAV3-P104L was observed in the GC, absent from the PM, and, in contrast to CAV3, rarely visible in the ER or LDs (Fig. 6, F and G; and Fig. S5). As expected for mutants unable to oligomerize, CAV1-P132L (Fig. 6, H and I) and CAV3-P104L (Fig. 6, L and M)

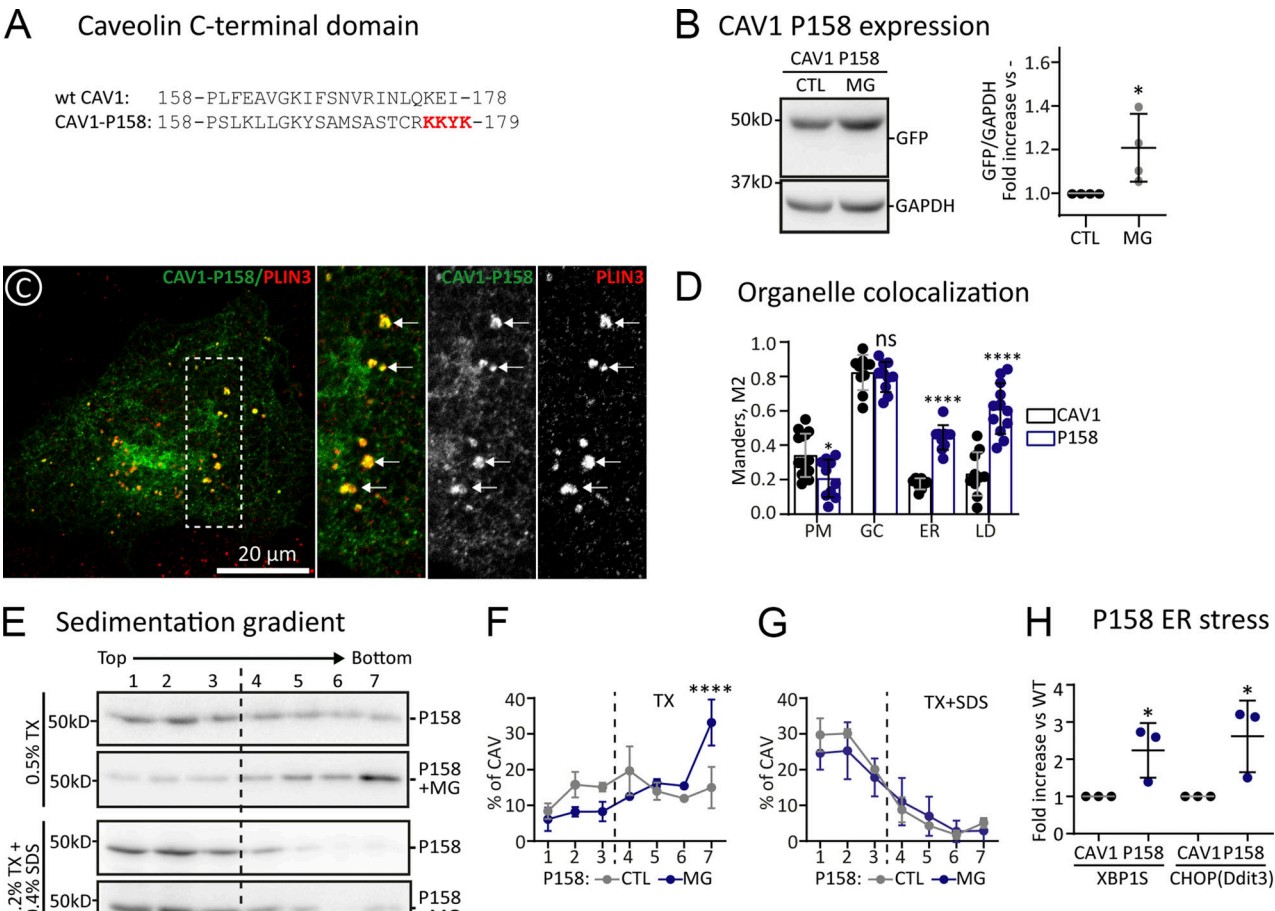

Figure 5. **Impaired arrival of CAV1 to late-GC avoids oligomerization. (A)** Amino acid sequence of the C-terminal domains of *wt* CAV1 and the CAV1-P158PfsX22 mutant (CAV1-P158). The newly generated ER-retrieval signal is indicated with red letters. **(B)** GFP-tagged CAV1-P158 was expressed for 3 h in CTL cells or in cells additionally treated with MG132. Protein levels were determined by densitometry as in Fig. 2 B (*n* = 3 independent experiments). **(C and D)** GFP-CAV1 and GFP-CAV1-P158 were expressed for 3 h, and their intracellular distribution analyzed by confocal microscopy. C shows representative images of the mutants and their co-localization with Plin-3 (LD marker). Additional images are included in Fig. S5. D shows the colocalization of CAV1-P158 with organellar markers as in Fig. 2 H (*n* ≥ 3 independent experiments and at least eight cells per condition). **(E–G)** CAV1-P158 was expressed for 3 h in CTL cells or in cells additionally treated with MG. Cells were solubilized with either TX or TX+SDS and fractionated in sucrose sedimentation gradients and protein distribution was analyzed as in Fig. 2 K with anti-GFP antibodies. E shows a representative result, and F and G show the relative distribution of the protein in each fraction of the gradient (*n* = 3 independent experiments). **(H)** CAV1 *wt* and CAV1-P158 were transfected in parallel and expressed for 16 h. The ER stress of cells was evaluated by measuring expression of XBP1 and DDIT3 by qPCR. Results are referred to the expression of these stress markers in cells transfected with CAV1 (*n* = 4 independent experiments). All graphs show means ± SD; ns, not significant; *P < 0.05, **P < 0.01, ***P < 0.001, ****P < 0.0001 in a two-tailed paired *t* test (B and H) or a two-way ANOVA test (D, F, and G). Scale bars are 20 µm. Source data are available for this figure: SourceData F5.

largely remain as low oligomeric species in TX sedimentation gradients. However, a pool of the mutants sedimented into high-molecular-weight species, a progressive process increasing when mutants were expressed for longer times (Fig. S5), likely reflecting its tendency to aggregate. Indeed, when expressed for 3 h in cells treated with MG132 the mutants, especially CAV3-P104L, formed high-molecular-weight aggregates that were insoluble in TX but efficiently solubilized in TX+SDS (Fig. 6, H–N). Intracellular accumulated CAV1-P132L and CAV3-P104L are toxic proteins for cells as both significantly activated the UPR as compared with *wt* caveolins (Fig. 6, K and O).

Therefore, these mutants demonstrate that oligomerization of caveolins is dispensable for the ER to GC transport but mandatory for GC to PM trafficking. Caveolin oligomerization occurs in late-GC, with mutants retained in the ER or early-GC remaining as low

oligomeric species. Despite active degradation, the mutants accumulate intracellularly and tend to form aggregates in the ER (CAV1-P158), GC (CAV1-P132L), or both (CAV3-P104L). Active degradation of intracellularly retained caveolin explains why the steady-state levels of these mutants are considerably lower when transfected for longer times and compared with *wt* proteins, which are highly stable when reaching the PM (Copeland et al., 2017; Galbiati et al., 1999, 2000; Kuga et al., 2011; Lee et al., 2002). The intracellular accumulation of caveolins causes cellular stress triggering the UPR.

**Impaired proteostasis of newly synthesized caveolins causes organellar damage**
Finally, we analyzed if an impaired degradation of caveolins is deleterious for cells by characterizing the K*R caveolin mutants.

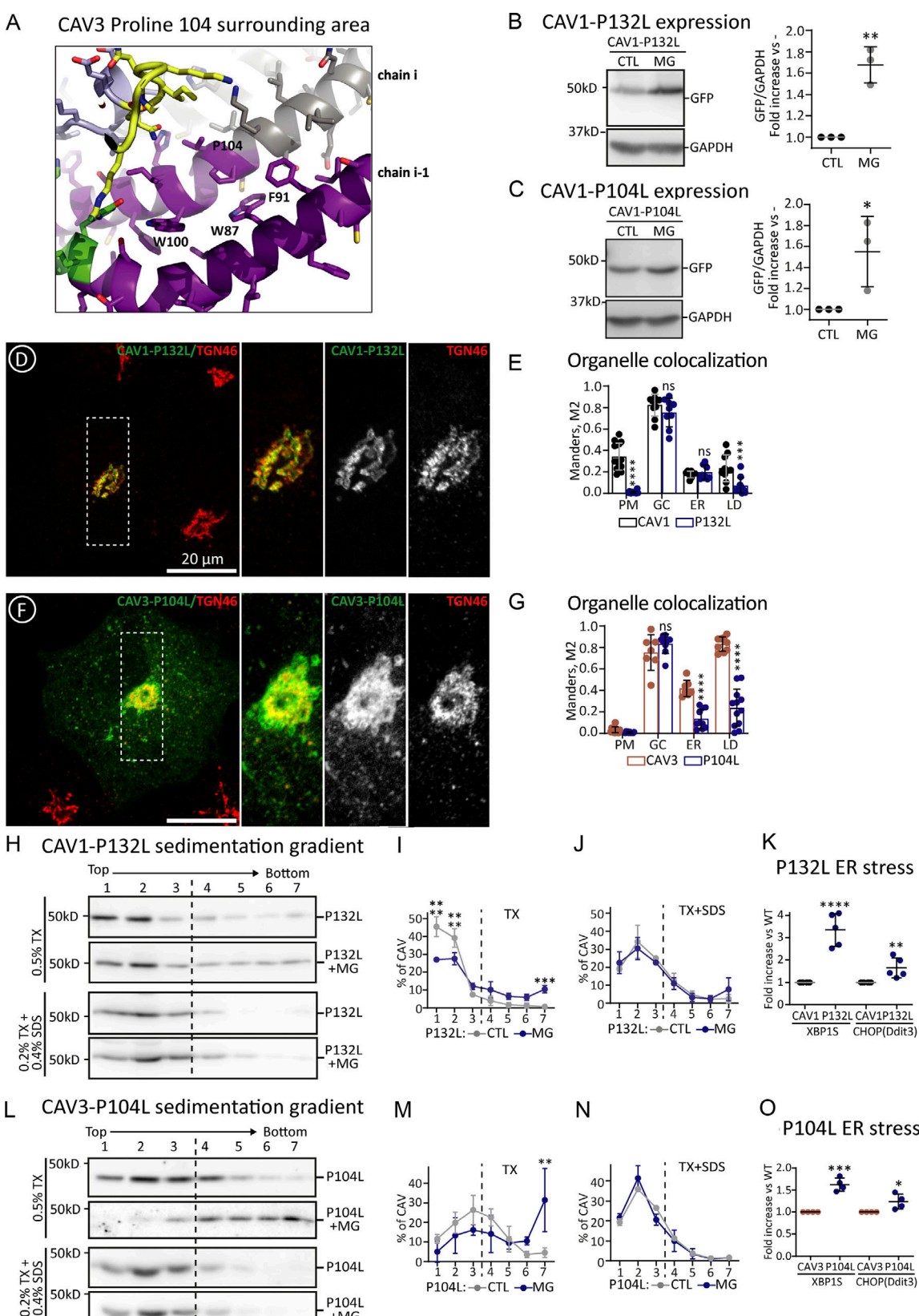

Figure 6. **Oligomerization defective caveolin mutants aggregate and cause ER stress. (A)** The scheme shows a detail of the position and interactions of the Pro 104 of human CAV3 within the oligomer predicted by AlphaFold2 (shown in ribbon diagram and colored according to the pLDDT score; Fig. 1 for additional details). **(B and C)** GFP-tagged CAV1-P132L (B) and CAV3-P104L (C) were expressed for 3 h in untreated cells or in cells additionally treated with MG132 and protein steady levels determined as in Fig. 2 B (n = 3 independent experiments). **(D–G)** The distribution of the mutants after 3 h was analyzed by

confocal microscopy. **(D and F)** show representative images of the mutants and their colocalization with TGN36 (GC marker). Additional images are included in Fig. S5. **(E and G)** show the colocalization of the mutants with organellar markers calculated as in Fig. 2 H ($n \geq 3$ independent experiments and at eight cells per condition). **(H–O)** Cells were transfected with GFP-tagged CAV1-P132L or CAV3-P104L and expressed in CTL cells or cells additionally treated with MG132. After 3 h cells were solubilized with either TX or TX+SDS and fractionated in sucrose sedimentation gradients as in Fig. 2, K and M. Protein distribution was quantified in equal volumes of each fraction by IB with anti-GFP antibodies. **(H and L)** shows a representative result and **(I, J, M, and N)** the relative distribution of each protein in each fraction of the gradient calculated by densitometry ($n = 3$ independent experiments). **(K and O)** CAV1 and CAV1-P132L or CAV3 and CAV3-P104L were transfected in parallel for 16 h and the ER stress evaluated by measuring expression of XBP1 and DDIT3 by qPCR. Results are referred to the expression of the stress markers in cells transfected with the *wt* caveolins ($n \geq 4$ independent experiments). All graphs show means ± SD; ns, not significant; *P < 0.05, **P < 0.01, ***P < 0.001, ****P < 0.0001 in a two-tailed paired *t* test (B and C, and stress in K and O) two-way ANOVA test (E and G) and sedimentation gradient in I, J, O, and P. Scale bars are 20 µm. Source data are available for this figure: SourceData F6.

When expressed for 16 h, the CAV1 K*R and CAV3 K*R mutants triggered the UPR (Fig. 7 A), which was especially significant for CAV1 K*R. Next, the effect of CAV1 K*R on cellular ultrastructure was analyzed by correlative light and electron microscopy (CLEM). Cells transfected overnight with either GFP-CAV1 or GFP-CAV1 K*R were first identified by light microscopy and then processed for electron microscopy (TEM). Serial TEM images were acquired at the regions of interest (ROIs) and fitted to the confocal z-stack using a combination of cellular landmarks and mitotracker (red signal, light microscopy, LM). As observed at earlier times of expression, GFP-CAV1 (Fig. 7 B) and GFP-CAV1 K*R (Fig. 7 C) showed distinct labeling patterns at the LM level. CAV1 showed dispersed labeling of puncta and relatively low labeling of the perinuclear area of the cell (including the GC) whereas CAV1 K*R showed a highly concentrated labeling of the putative GC. This was confirmed by correlative EM; CAV1 showed low labeling in the GC (Fig. 7, D–F) but CAV1 K*R was highly concentrated in abnormal perinuclear membranous structures (Fig. 7, G–I). In contrast to the labeling of typical GC cisternae in cells expressing CAV1 (Fig. 7 F, insert), the morphology of the putative GC in CAV1 K*R expressing cells was characterized by an abundance of small vesicular structures (Fig. 7 I, insert), suggesting a dramatic effect of the abnormal intracellular accumulation of caveolins on GC structure and function.

## Discussion

Our model cell system of synchronized caveolin expression followed by analyses at very early time points has identified several novel proteostasis mechanisms differentially regulating trafficking and steady-state levels of caveolin family members, isoforms, and mutants (Fig. 8).

First, in contrast to the current view, we demonstrate that caveolins are transported from the ER to the GC as monomers or small oligomeric species, and 11-mer (8S) rings are only assembled beyond cis-GC. These dynamics are (i) supported by our previous description that antibodies against the signature motif, which is hidden within the rings, still recognize the cis-GC pool of caveolin but not oligomers in the PM (Hayer et al., 2010a; Luetterforst et al., 1999; Pol et al., 2005); (ii) predicted by our in silico structural analysis showing that residues binding COPII are buried and inaccessible once the rings are formed; (iii) suggested here by the observation that CAV1-P132L is exported from the ER to the GC as efficiently as CAV1 and thus, oligomerization is dispensable for this step; (iv) demonstrated by the fact that newly synthesized CAV3, although residing in the ER for longer times, does not spontaneously oligomerize and remains in a monomeric or low oligomeric state; and (v) corroborated by CAV1-P158 that, although containing an intact oligomerization domain, organizes in low oligomeric species because it is unable to reach the late-GC. This new model is also consistent with earlier studies of CAV2; monomeric CAV2 was shown to be present in the GC in cells lacking CAV1 but to form oligomers and redistribute to the PM upon CAV1 expression (Sowa et al., 2003). Intriguingly, in contrast to the prediction of AlphaFold and the oligomerization occurring when expressed to high levels in bacteria (Walser et al., 2012), CAV2 remains in a low oligomeric state in the GC. We speculate that in the absence of CAV1, the CAV2 protein is unable to reach the late-GC, an obligatory site for caveolin oligomerization. If correct, the oligomerization of caveolin may provide new insights into the specific lipid environment of the late-GC.

Second, our study reveals a previously undescribed complexity in the early proteostasis of caveolins (Fig. 8). Failures in these sequential steps lead to the aggregation of caveolins. Protein aggregation, triggered by perturbations in protein homeostasis, has been linked to the onset of more than 40 human diseases including aging, diabetes, and neurodegeneration (Balchin et al., 2016; Hipp et al., 2019). In contrast to 8S oligomers, caveolin aggregates can be biochemically defined by being TX-insoluble but TX+SDS-soluble (Copeland et al., 2017). Formation of high-molecular-weight TX-insoluble non-functional complexes has been consistently reported for many CAV1 and CAV3 mutants (Copeland et al., 2017; Galbiati et al., 1999; Han et al., 2016a; Lee et al., 2002; Sotgia et al., 2003). Here, by comparison with those mutants, we have observed that newly synthesized *wt* caveolins are also prone to aggregation, for example, when the proteasomal machinery is compromised. The propensity of purified caveolins to form protein aggregates has also been demonstrated in artificial lipid membranes (Zhang et al., 2021). In our experimental systems, these dynamics are particularly well-illustrated for newly synthesized CAV3, which is actively maintained in a low oligomeric state in the ER by robust degradation but rapidly forming aggregates when proteasomal activity is inhibited.

Caveolin aggregation caused by disrupted proteostasis has two consequences. Aggregation leads to trafficking defects as illustrated using degradation-insensitive mutants and by the observed formation of caveolin complexes retained along the biosynthetic pathway. The capacity of aggregates to impair caveolin trafficking is illustrated by the dominant negative

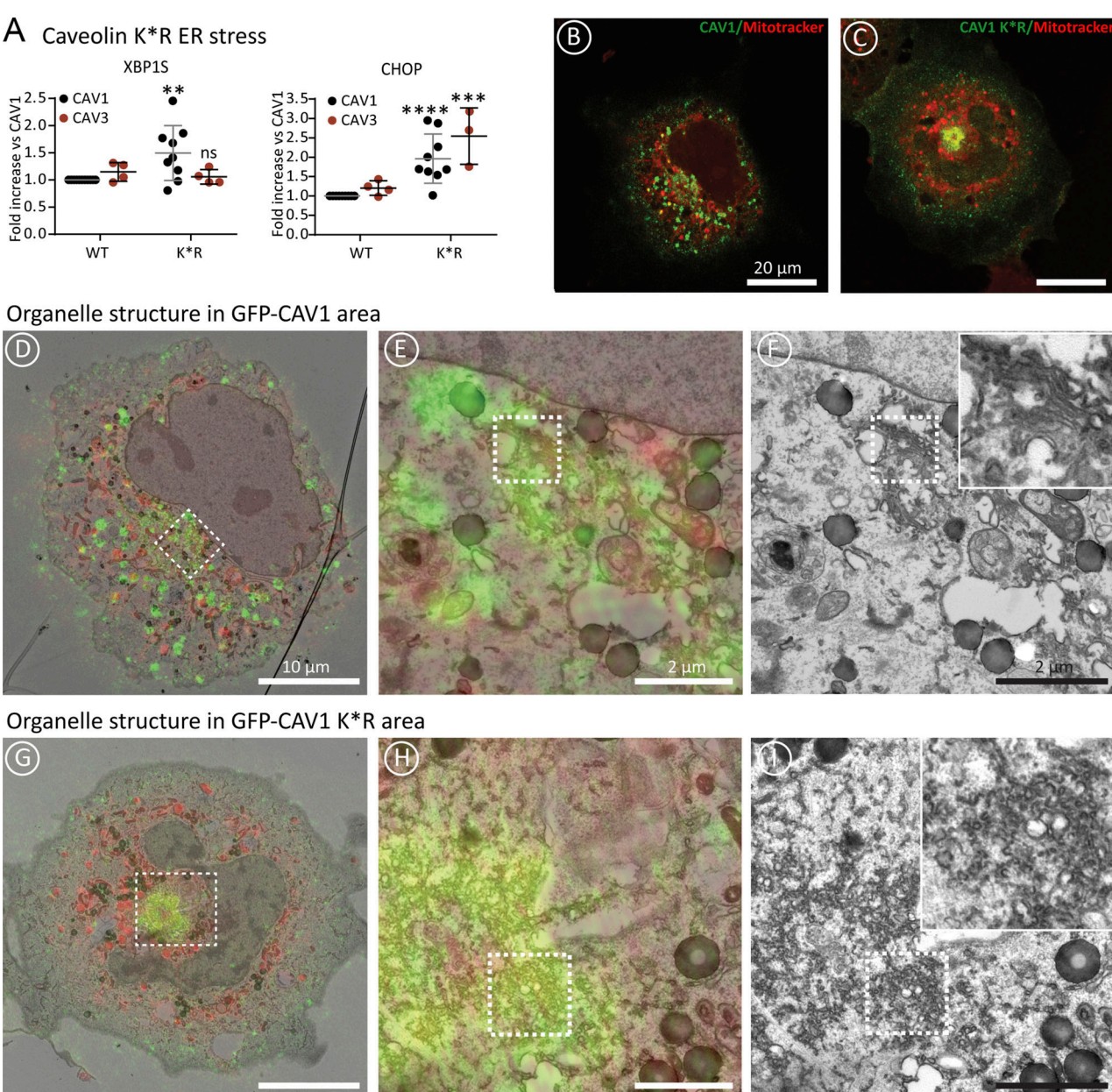

**Figure 7.** **Impaired proteostasis of newly-synthesized caveolins causes organellar damage. (A)** CAV1 and CAV1 K*R or CAV3 and CAV3 K*R were transfected in parallel for 16 h and the ER stress evaluated by measuring expression of XBP1 and DDIT3 by qPCR. Results are referred to the expression of the stress markers in cells transfected with the *wt* caveolins (n ≥ 3 independent experiments). The graphs show means ± SD; ns, not significant; *P < 0.05, **P < 0.01, ***P < 0.001, ****P < 0.0001 in a two-way ANOVA test. **(B–I)** Cells transfected with GFP-CAV1 (B and D–F) or GFP-CAV1 K*R (C and G–I) were incubated with Mitotracker (red) and fixed for CLEM. The figure includes single confocal slices (B for CAV1 and C for K*R) and the overlays of the fluorescently labeled images over the corresponding EM sections (D and E for CAV1 and G and H for K*R). The squares indicate the areas selected for high magnification panels shown next. GFP-CAV1 shows labeling of puncta throughout the cell (D) whereas GFP-CAV1 K*R is very concentrated in the perinuclear area (G). GFP-CAV1 labels typical Golgi cisternae (E and F). In contrast, high labeling for GFP-CAV1 K*R is associated with abnormal small vesicles/tubules clustered in the perinuclear area of the cell, demonstrating the disruption of Golgi complex structure (H and I). Bars for low- and high-magnification panels are indicated.

effect of mutated caveolins that, prone to aggregate, intracellularly sequester *wt* caveolins and reduce caveola formation (Copeland et al., 2017; Galbiati et al., 1999; Lee et al., 2002; Pol et al., 2001; Roy et al., 1999). Furthermore, intracellularly retained caveolins trigger ER stress and have the potential to disrupt organelles, as exemplified by the effect of the non-degraded CAV1 K*R mutants on the GC. Therefore, the complex proteostatic mechanisms identified here ensure forward

trafficking and avoid cell damage until caveolin arrives at the late-GC, where functional oligomers are assembled and rapidly exported to the PM.

Due to the well-documented harmful effects of *wt* caveolin when expressed at high levels (Hanson et al., 2013), these proteostatic mechanisms might be particularly relevant in situations where these proteins are produced in high amounts to meet specific needs such as the formation of the zebrafish

## Caveolin early proteostasis and oligomeric state.

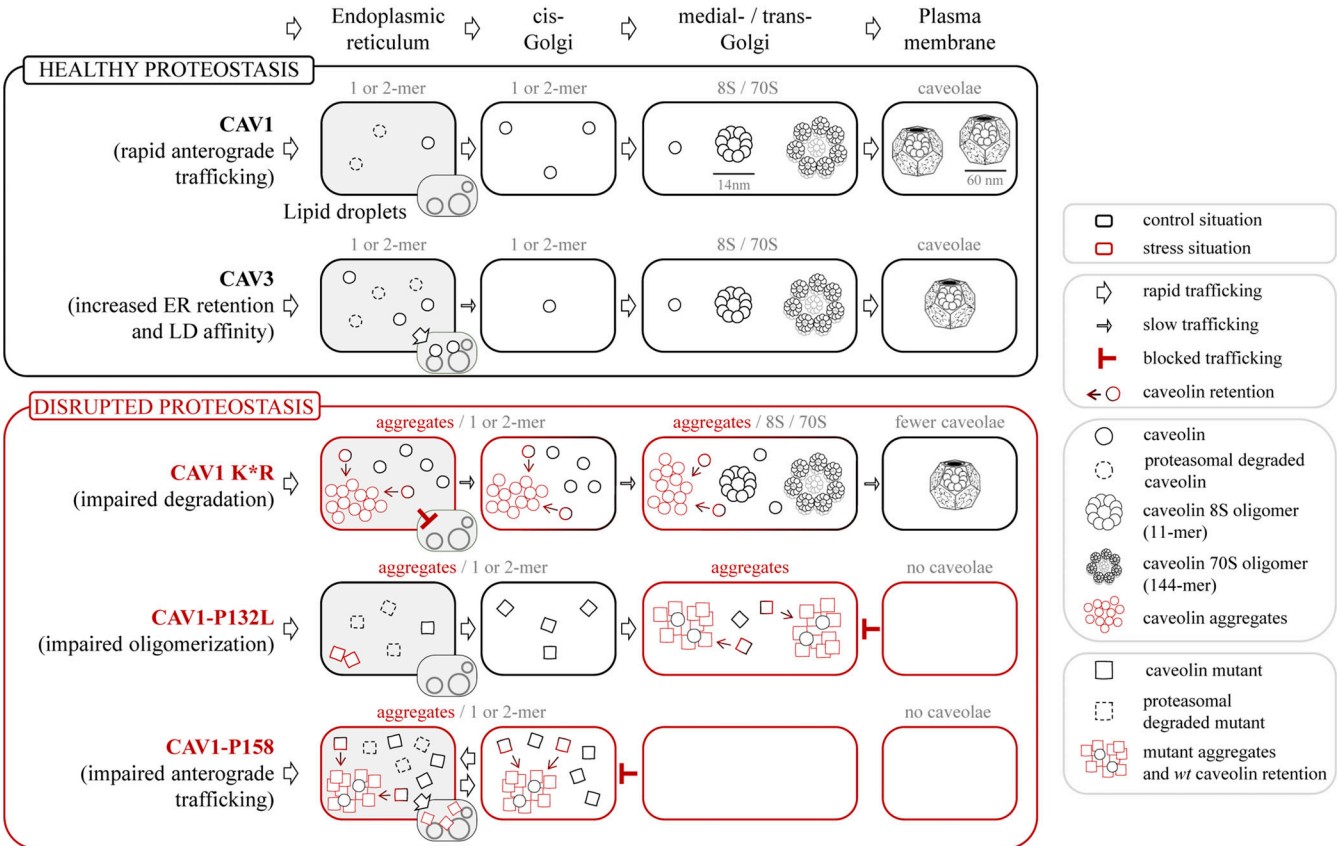

Figure 8.  **Early proteostasis of caveolins and disruption by pathogenic mutants.** Caveolin protomers (black circles) are co-translationally inserted into the ER through the central hydrophobic domain in a signal recognition particle (SRP)-dependent manner. CAV1 is rapidly transported to the GC in COPII vesicles. CAV3 is retained for longer times in the ER and LDs than CAV1, with fewer CAV3 protomers moving to the GC. Because the COPII binding residues of caveolins are completely hidden within large oligomeric complexes, only monomeric or small oligomers of caveolins can be transported from the ER to the cis-GC. Caveolins are submitted to a robust degradation in the ER (dashed circles), a process that prevents the tendency of caveolins to aggregate (see below). Once in the GC, probably in medial- or late-GC, and likely by finding a specific lipid environment, caveolins oligomerize, initially into 8S oligomers (11 protomers) and next into 70S oligomers (~144 protomers), which are rapidly transported to the PM to assemble mature caveolae. Active proteasomal degradation is needed for efficient caveolin trafficking, likely by avoiding the aggregation of caveolins, a process also observed in artificial membranes and probably masking key sorting motifs. Degradation-insensitive mutants (K*R, red circles) have a marked tendency to form aggregates and display trafficking defects; slow transport to the PM for CAV1 K*R and to LDs in the case of CAV3 K*R. Similarly, *wt* caveolins display trafficking defects when the proteasome is chemically inhibited. The K*R mutants trigger ER stress and disrupt secretory pathway organelles. Caveolin pathogenic mutants unable to oligomerize, such as CAV1-P132L (red squares), accumulate in the GC. Although subjected to degradation, these mutants also progressively form aggregates and trigger ER stress. The aggregates formed by mutants have the capacity to retain *wt* caveolins (black to red circles) and reduce caveola formation. Equivalent mutants of CAV3, such as CAV3-P104L, also accumulate and aggregate in the ER and trigger stress. In support of this model, caveolin mutants unable to reach the late-GC, such as CAV1-P158, fail to form functional oligomers, aggregate, and cause stress. When early proteostatic mechanisms fail, for example in the environment of unhealthy or senescent cells, instead of forming PM caveolae, *wt* caveolins could be toxic proteins and additional stressors.

notochord (Nixon et al., 2007), adipogenesis (Scherer et al., 1994), or muscle differentiation (Song et al., 1996). If early proteostasis fails, for example in the environment of unhealthy or senescent cells, caveolins could be toxic proteins and additional stressors, explaining why caveolin depletion ameliorates the progression of aging (Roitenberg et al., 2018), diabetes (Zeng et al., 2018), and neurodegeneration (Trushina et al., 2014). The possibility that caveolin aggregates have additional effects on cellular homeostasis by sequestering other proteins or lipids deserves additional analyses.

Third, we demonstrate striking differences in the early trafficking of CAV1 and CAV3, with CAV1 rapidly transported to the PM but CAV3 retained in the ER for longer periods. Because our data strongly suggest that oligomerization occurs in the late-GC, a change in folding/oligomerization would not explain the retention of CAV3 in the ER. Further, from our in silico analysis and from previous studies of CAV1 and CAV3 expressed in bacteria (Walser et al., 2012), we have no indications that differences in oligomerization of CAV3 as compared with CAV1 mediate a differential trafficking. Instead, our results suggest that an N-terminal alpha-helix retains CAV3 in the ER and less protein reaches the oligomerization sites in the late-GC. Whether the CAV3 N-terminal alpha helix, which is dispensable for trafficking to the PM or LDs (Luetterforst et al., 1999; Pol et al., 2001), interacts with lipids or proteins in the ER or it is functioning as an ER-retention signal by modulating the affinity of the proximal COPII binding domain (Ma et al., 2017) deserves further studies.

Newly synthesized CAV1 and CAV3 also differ in their affinity for LDs. In contrast to CAV1, CAV1-P158 has a high affinity for LDs, suggesting that the new amino acids enclose an LD-targeting motif. Even when retained in the ER for longer times, like in the case of CAV1 N-CAV3, CAV1 demonstrates a low capacity for lateral trafficking into LDs, suggesting that the CAV1 pool commonly observed on LDs is trafficking from the PM (Bersuker et al., 2018; Bosch et al., 2020b; Pol et al., 2005). This differential trafficking could be synchronized with the transcriptional regulation of caveolins. CAV1 is transcriptionally regulated by cholesterol (Hailstones et al., 1998) and the presence of cholesterol accelerates the transport to the PM (Hayer et al., 2010a; Pol et al., 2005), suggesting that CAV1 could function as a cholesterol sensor and transporter (Pol et al., 2020). In contrast, the CAV3 gene is under the control of the retinoic acid receptor (RAR)-related orphan receptor alpha (RORα; Lau et al., 2004), a transcription factor involved in fatty acid metabolism, activation of mitochondria, and regulation of genes involved in circadian rhythms. While the precise role of caveolins on LDs is not understood (Parton et al., 2020; Pol et al., 2020), the shorter lifetime of newly synthesized CAV3 and their high affinity for LDs could be part of a bioenergetic program (Bosch et al., 2020a) evolved specifically for cardiac and skeletal muscles to rapidly but transiently respond to fluctuations in fatty acid availability (Gonzalez Coraspe et al., 2018). To our knowledge, this is the first time that striking functional differences between CAV1 and CAV3 have been observed and might explain the existence of two different caveolins with, until now, an identical ability to form caveolae (Kirkham et al., 2008; Pol et al., 2020).

In conclusion, these studies emphasize the essential role of proteostatic mechanisms in the life cycle of caveolins and identify distinct trafficking properties of the family members. The formation of the caveolin oligomeric disc must be highly regulated in cells with aberrant formation of this unique structure having profound effects on cell physiology. Perturbation of these mechanisms leads to precocious aggregation of caveolins, trafficking defects, and organelle stress, likely favoring disease progression. Interestingly, specialized mechanisms also exist for specific degradation of mature caveolin oligomers in post-GC endosomal compartments (Hayer et al., 2010b; Ritz et al., 2011), emphasizing the precise regulatory mechanisms that have evolved to control caveolin levels at various stages of its trafficking itinerary, both before and after reaching the PM. Identification of these early proteostatic mechanisms provides new insights into the poorly understood roles of intracellular caveolins and importantly, into the pathogenic potential of caveolins when there is a perturbation of the biosynthetic steps mandatory for the assembly of oligomeric rings and formation of mature caveolae.

## Materials and methods

### Plasmids
Mouse pEGFP-CAV1, pEGFP-CAV3, pEGFP-CAV1-P132L, and pEGFP-CAV3-P104L were purchased from Genscript. pEGFP-CAV2 was obtained by PCR from pCMV6-MycFlag-CAV2 (#MR201276; Origene) using primers containing BglII and HindIII sites and subcloned into pEGFP. pEGFP-CAV1 K*R was obtained from Addgene (#27766). cDNA of CAV3 K*R was acquired from GenScript and subcloned into pEGFP using BglII and HindIII sites. pEGFP-CAV1β was generated by PCR using primers starting at MADEVT and containing BsrgI and BamHI sites. pEGFP-CAV1-NCAV3 was designed by exchanging the hypervariable region of CAV1β (MADEVTEKQVYDAHT) with the N-terminal region of CAV3 (MMTEEHTDLEARIIKDIHC), while the opposite exchange was made for pEGFP-CAV3-NCAV1. Both sites were swapped after BsrglI/BamHI digestion. pEGFP-CAV1-P158 was designed by replacing the C terminus of CAV1 (158-PLFEAVGKIFSNVRINLQKEI) with the C terminus of the CAV1 P158PfsX22 frameshift mutation (158-PSLKLLGKYSAMSASTCRKKYK). Both sites were swapped after an EcoRI/XmaI digestion. pECFP-VAP-A was kindly provided by Dr. Carles Enrich (University of Barcelona, Barcelona, Spain) and subcloned into a pEGFP vector using BsrgI and EcoRI sites. pEGFP-Syntaxin 6 was kindly provided by Dr. Jeffrey Pessin (Albert Einstein College of Medicine, Bronx, NY, USA). pEGFP-ALDI was previously cloned (Turro et al., 2006). pEG-MycFlag-CAV1, pEG-MycFlag-CAV2, and pEG-MycFlag-CAV3 were obtained from pEGFP-CAVs vectors replacing EGFP with MycFlag tag.

### Protein structural prediction, modeling, and visualization
The structural predictions of human caveolin paralogs were performed using the AlphaFold2 neural network (Jumper et al., 2021), implemented within the ColabFold pipeline (Mirdita et al., 2022), using default settings and MMseqs2 for multiple sequence alignments (Mirdita et al., 2019). Sequence accession numbers are provided in Table S1. Monomeric predictions were performed using both AlphaFold2-ptm and AlphaFold2-multimer-v2 and compared with the cryoEM structure of CAV1 monomers. AlphaFold2-ptm was used for the modeling of larger homo-oligomers with a varying number of subunits and disc-like structures for all the caveolins. Homopentameric CAV1, CAV2, and CAV3 were focused on due to the high similarity of these models to pentamers derived from the CAV1 cryoEM structure, and their inter-subunit interactions were analyzed from i−2 to i+2 ring. Surface hydrophobicity images were mapped using Protein-Sol Patches server (Hebditch and Warwicker, 2019) and structures were rendered with Pymol (https://pymol.org/2/; Schrodinger).

### Cells and reagents
COS1 (ATCC CRL-1650) and C2C12 (ATCC CRL-1772) were cultured in Dulbecco's modified Eagle's medium (DMEM; Biological Industries) supplemented with 10% v/v fetal bovine serum (Biological Industries) and 4 mM L-glutamine, 1 mM pyruvate (Sigma-Aldrich), 50 U/ml penicillin, 50 μg/ml streptomycin, and non-essential amino-acids (Biological Industries). Cycloheximide (CHX, C7698) was purchased from Sigma-Aldrich. MG132 (MG) was purchased from Merck (474790).

### Cell transfection and treatment
GenJet Plus (Signagen) and Lipofectamine LTX (Invitrogen) reagents were used to transfect COS1 and C2C12 cells respectively following the manufacturer's instructions. Before the addition of

the lipid–DNA complex, 50 μg/ml CHX was added. After 6 h of incubation, washout of CHX was performed by three washes with complete media and gentle tapping, and synchronized protein expression was allowed for 3, 16, or 24 h in normal media. In some experiments, the proteins were expressed in a media additionally supplemented with 50 μM MG132, 1.1 mM OA, or 4 mM cholesterol.

## Total cell lysates
Transfected cells were washed twice with cold PBS and gently scrapped with 80 μl of lysis buffer (50 mM Tris HCl pH 7.5, 150 mM NaCl, and 5 mM EDTA, and protease and phosphatase inhibitors) with 0.1% Triton X-100 (T8787; Sigma-Aldrich). Collected samples were completely disrupted by sonication for 20 s at 30 Amps, and protein concentration was quantified using the Bio-Rad protein assay kit (500-0006; Bio-Rad). 30–60 μg of protein were loaded on an SDS-page gel.

## 1% triton solubilization assay
The protocol was adapted from (Schlegel et al., 1999) and performed on ice. Transfected cells were washed two times with chilled PBS and solubilized for 35 min without agitation with 300 μl of cooled lysis buffer with 1% Triton X-100. The soluble fraction was collected by decantation and pipetting. The insoluble fraction was collected by scraping cells with 300 μl of SDS lysis buffer (2% SDS, 1% Tris-HCl pH 6.4) and further disrupted by passing the lysate through a 25-G needle 10 times. 30 μl of all fractions were loaded on an SDS-page gel.

## Sucrose sedimentation gradient
The protocol was adapted from (Hayer et al., 2010a). $1 \times 10^6$ cells grown in 60-mm well plates were washed two times with PBS, gently scrapped in 500 μl PBS, and centrifuged at 800 G for 4 min at RT. Cells were permeabilized in 200 μl of either 0.5% Triton X-100 or 0.2% Triton X-100 and 0.4% sodium dodecyl sulfate lysis buffer for 20 min and centrifuged for 5 min at 1,100 G to discard insolubilized cells. 200 μl of supernatant were loaded in the upper phase of a sucrose step gradient of 500 μl of 40%, 30%, 20%, and 10% sucrose in lysis buffer, in a polypropylene centrifuge tube (347357; Beckman Coulter). Gradient tubes were then ultracentrifuged (Sorval MX 150; Thermo Fisher Scientific) at 55,000 rpm for 4 h 15 min at 4°C (S55S; Hitachi rotor). Seven fractions of 310 μl were collected from top to bottom, and 30 μl of all fractions were loaded on an SDS-page gel.

## Sucrose flotation gradient
Two 100-mm plates with $3 \times 10^6$ transfected cells treated with 1.1 mM oleic acid for 3 h after CHX washout were used for each condition. Cell's lysates were obtained by gently scraping in 300 μl of lysis buffer followed by cavitation at 55 Bar for 15 min at 4°C. Samples were passed through a 22-G needle 25 times and centrifuged at 1,500 G for 5 min at 4°C. 500 μl of the supernatant fraction was mixed with 500 μl of 65% sucrose and placed at the bottom of a step gradient of 200 μl of 30%, 25%, 20%, 15%, 10%, and 5% (w/v) sucrose in lysis buffer and ultracentrifuged (Sorval MX 150; Thermo Fisher Scientific) at 50,000 rpm for 3 h at 4°C

(S55S; Hitachi rotor). Five fractions of 300 μl were collected using a tube slicer from top to bottom and 700 μl were collected from the bottom phase. 30 μl of all fractions were loaded on an SDS-PAGE gel.

## SDS-PAGE and immunoblotting
Protein samples were loaded onto a polyacrylamide gel using a Bio-rad Mini-Protean II electrophoresis cell and transferred to a nitrocellulose membrane. Membranes were blocked in 5% non-fat dry milk in TBS-T for 1 h and incubated with the primary antibodies anti-GFP (1:5,000, rabbit, polyclonal; ab290; Abcam), anti-GAPDH (1:5,000, goat, polyclonal; A00191; Genescript), anti-CAV1 (1/3,000, rabbit, polyclonal; 610060; BD Biosciences), and anti-Plin3 (1:300, guinea pig, polyclonal; GP32; ProgenMembranes were washed and incubated with peroxidase-conjugated secondary antibodies (1:3,000; Bio-Rad) and detected with ECL (Biological Industries). Immunoblots were visualized using the ImageQuant LAS4000 (GE Healthcare) digital imager. Immunoblot quantification was performed using the ImageJ software (National Institutes of Health). For measuring protein steady-state levels, band intensity was quantified by densitometry, corrected with respect to the GAPDH levels, and referred to the GFP levels of the *wt* protein. In triton solubilization assays, both soluble and insoluble fractions were quantified by densitometry. Both fractions were represented as the percentage of the total amount of protein. Insoluble fraction increase is also plotted and referred to as the *wt* protein. Fractions of both floatation and sedimentation gradients were quantified by densitometry and represented as the percentage of the sum of intensities.

## Fluorescence microscopy
Cells were grown in 10 mm glass coverslips, fixed for 1 h with 4% PFA, permeabilized in 0.15% Triton X-100, and blocked with 1% BSA (A7906; Sigma-Aldrich). Cells were labeled by the incubation with primary antibodies: anti-Cavin1 (1:100; 18892-1-AP; Proteintech), anti-Calreticulin (1:75, rabbit, polyclonal; ab2907; Abcam), anti-Plin3 (1:700; GP32; Progen), and anti-TGN46 (1:300, sheep, polyclonal; AHP500GT; Bio-Rad) for 1 h. Labeled cells were stained by incubation with secondary antibodies conjugated with Alexa Fluor dyes (Invitrogen) for 45 min. Coverslips were finally mounted with Mowiol mounting media (475904; Calbiochem).

Images were taken with a Zeiss LSM880 laser scanning spectral confocal microscope equipped with an AxioObserver Z1 inverted microscope, objective 63X Oil with 1.4 NA, and three photomultiplier tube (PMT) for fluorescence, one of them is a GaAsP PMT detector and one Transmitted Light detector, at room temperature. The acquisition software used is Zeiss Zen software with advanced programming experiments.

## Image analysis
Images were analyzed using the Adobe Photoshop CS3 software (Adobe Systems Inc) and FIJI-ImageJ (1.52i; National Institutes of Health). Organelle colocalization was measured by the use of JACoP plugin (Bolte and Cordelières, 2006) in ImageJ/FIJI, which was automated in a customized macro to calculate the Mander's M2 overlap coefficient.

## Correlative light and electron microscopy

COS1 cells grown on glass bottom gridded dishes (P35G-1.5-14-C-GRD; Mattek) were transfected overnight (16 h) with GFP-CAV1 or GFP-CAV1K*R. Prior to fixation, cells were incubated with Mitotracker CMXRos (M-7512; Invitrogen) according to the manufacturer's instructions. Cells were then fixed in 0.2% glutaraldehyde/4% paraformaldehyde for 1 h at RT in PBS, washed with PBS, and then imaged in a Leica LSM 880 confocal microscope (Leica Microsystems). A z-stack was prepared for cells of interest and the coordinates were recorded for later identification of cells of interest. The cells were processed for TEM as described previously (Takasato et al., 2015). Sections of the ROIs were prepared and compared with individual slices from the confocal stack. ROIs were identified and re-registered using a combination of cellular landmarks as observed by EM (mitochondria, lysosomes, Golgi) plus the fluorescent signals in the confocal stack (GFP, Mitotracker; note that Mitotracker labels both mitochondria and endolysosomes under these labeling conditions). Images were fitted using Adobe Photoshop with Puppet Warping, which enables alignment of LM and EM images in view of the differences in the cellular dimensions of intact cells before processing (LM) and the dimensions of the same cells after dehydration and resin embedding (EM).

## Gene expression by quantitative PCR (qPCR)

Cells in 35-mm well plates were washed two times with PBS and RNA was isolated using the EZ-10 DNAway RNA Mini-Preps kit (BS88136; Biobasic) according to manufacturing instructions. cDNA was synthesized from 1 µg RNA using the High-Capacity cDNA Reverse Transcription Kit (Applied Bioscience, Thermo Fisher Scientific). qRT-PCR was performed using SYBRGreen-based technology GoTaq qPCR master-mix (A600A; Promega) and LightCycler 96 Real-time PCR system (Roche). The following primers were used for the qPCR: 36B4: forward, 5′-CAGCAAGTGGGAAGGTGTAATCC-3′, and reverse, 5′-CCCATTCTATCATCAACGGGTACAA-3′. XBP1S: forward, 5′-GGTCTGCTGAGTCCGCAGCAGG-3′, and reverse, 5′-AGTTCATTAATGGCTTCCAGCT-3′. CHOP (Ddit3): forward, 5′-CCTCCTGGAAATGAAGAGGAAGAA-3′, and reverse, 5′-GCTAGCTGTGCCACTTTCCT-3′, and reverse. Analysis of gene expression was performed using the ΔΔCt method using 36B4 as a housekeeping gene.

## Statistical analysis

All data shown in the graphs are the mean ± SD. Statistical significance was determined using paired $t$ test, one-way and two-way analysis of variance (ANOVA) multiple comparisons test as specified in figure legends (not significant [ns], *$P < 0.05$, **$P < 0.01$, ***$P < 0.001$, and ****$P < 0.0001$). The number of independent experiments ($n$) is indicated in each figure legend. Data distribution was assumed to be normal, but this was not formally tested.

## Online supplemental material

Fig. S1 shows AlphaFold2 predictions of pentameric homo-oligomers of human CAV1, CAV2, and CAV3; supplemental for Fig. 1. Fig. S2 shows differential early transport of newly synthesized caveolins; supplemental for Fig. 2. Fig. S3 shows proteasomal degradation determines the trafficking of newly synthesized caveolins; supplemental for Fig. 3. Fig. S4 shows that the caveolin N-terminus determines the localization of newly synthesized caveolins; supplemental for Fig. 4. Fig. S5 shows intracellular distribution of caveolin pathogenic mutants; supplemental for Figs. 5 and 6. Table S1 lists key resources.

## Acknowledgments

A. Pol and M. Bosch are supported by I+D+i RETOS INVESTIGACIÓN from the Ministerio de Ciencia e Investigación (MICINN, PID2021-127043OB-I00) and the CERCA Programme/Generalitat de Catalunya. A. Pol is supported by the H2020-MSCA-ITN-2018, 953489 (ENDOCONNECT). R.G. Parton is supported by the National Health and Medical Research Council (NHMRC) of Australia (program grant, APP1037320) and by an Australian Research Council (ARC) Laureate Fellowship. B.M. Collins is supported by an NHMRC Senior Research Fellowship (APP1136021) and an ARC Discovery Project Grant (APP1181135). F. Morales-Paytuví is supported by the FPI program (MINECO, BFU2015-66401-R). R.G. Parton is a Laureate Fellow of the Australian Research. We thank Dr. Dominic Hunter and the members of the Parton, Collins, and Pol groups for critical reading of the manuscript. We would like to thank Melanie Ohi and Anne Kenworthy for providing the CAV1 structural coordinates prior to public release. We would also like to acknowledge and thank Milot Mirdita, Martin Steinegger, and the ColabFold team for making their pipeline available for public use, and Drs. Vikas Tillu and Michael Healy for discussion of structural modeling. Council. A. Pol and R.G. Parton are funded by the European Union (ERC-2022-SYG, 101071784, DRIMMS). Views and opinions expressed are however those of the author(s) only and do not necessarily reflect those of the European Union or the European Research Council. Neither the European Union nor the granting authority can be held responsible for them.

Author contributions: Conceptualization: R.G. Parton and A. Pol. Methodology: F. Morales-Paytuví, C. Ruiz-Mirapeix, M. Bosch, B.M. Collins, and A. Fajardo. Formal analysis: M. Bosch, R.G. Parton, B.M. Collins, and A. Pol. Investigation: F. Morales-Paytuví, A. Fajardo, F. Tebar, J. Rae, and C. Ruiz-Mirapeix. Resources and supervision: R.G. Parton, and A. Pol. Data curation: F. Morales-Paytuví, M. Bosch, R.G. Parton, and A. Pol. Writing (original draft): A. Pol. Writing (review and editing): R.G. Parton, B.M. Collins, and A. Pol. Visualization: B.M. Collins, R.G. Parton, and A. Pol. Funding acquisition: R.G. Parton and A. Pol.

Disclosures: The authors declare no competing interests exist.

Submitted: 7 April 2022

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

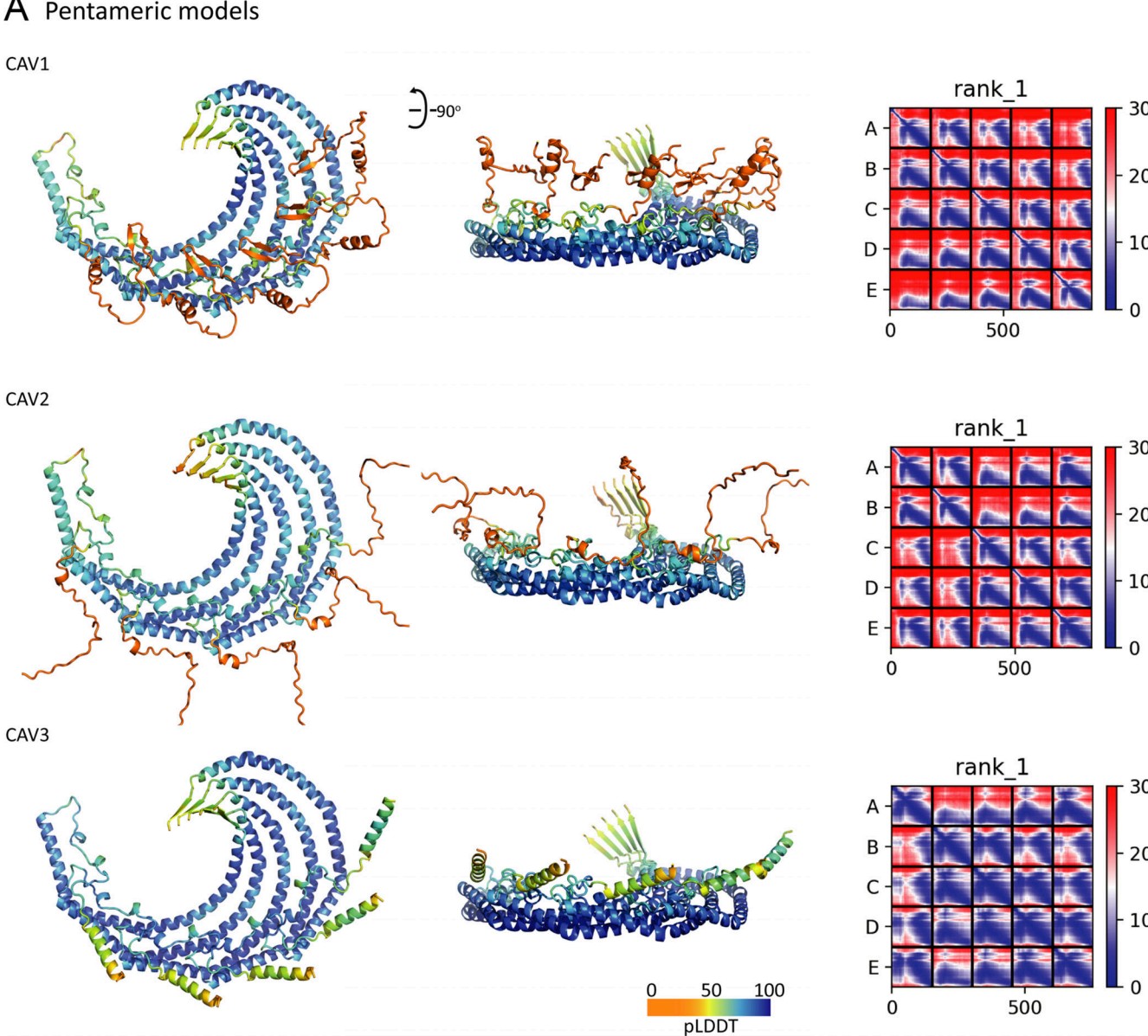

Figure S1. **AlphaFold2 predictions of pentameric homooligomers of human CAV1, CAV2, and CAV3.** Supplemental for Fig. 1. **(A)** The top-ranking structures of homo-pentameric models of human CAV1, CAV2, and CAV3 predicted by AlphaFold2 are shown in the ribbon diagram and colored according to the pLDDT scores. The right-hand panels show the plots of the Predicted Alignment Error (PAE) for each top-ranking model. There is a strong degree of correlation between the five chains in each structure indicating their physical association with each other.

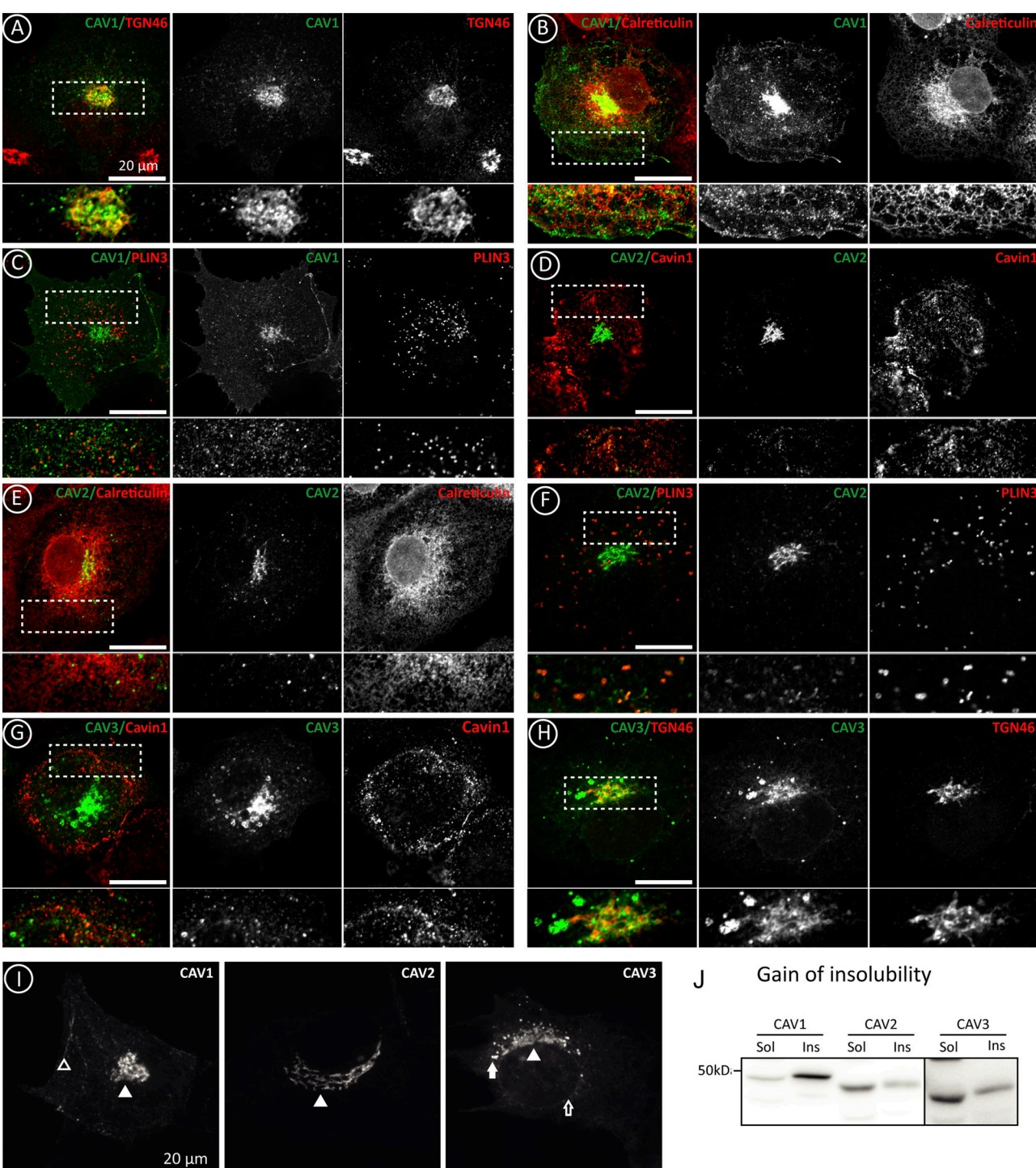

Figure S2.   **Differential early transport of newly synthesized caveolins.** Supplemental for Fig. 2. **(A–H)** Colocalization of GFP-CAV1, CAV2, and CAV3 with antibodies recognizing Cavin-1 (PM caveolae), TGN46 (GC), Calreticulin (ER), and Plin-3 (LDs) and analyzed by confocal microscopy. **(I)** Distribution of GFP-caveolins in C2C12 myoblasts after 3 h. The location of caveolin in the PM (open arrowhead), GC (arrowhead), ER (open arrow), and LDs (arrow) is indicated. **(J)** Gain of the insolubility of caveolins in C2C12 cells after a 3 h expression as explained in Fig. 2 I. Source data are available for this figure: SourceData FS2.

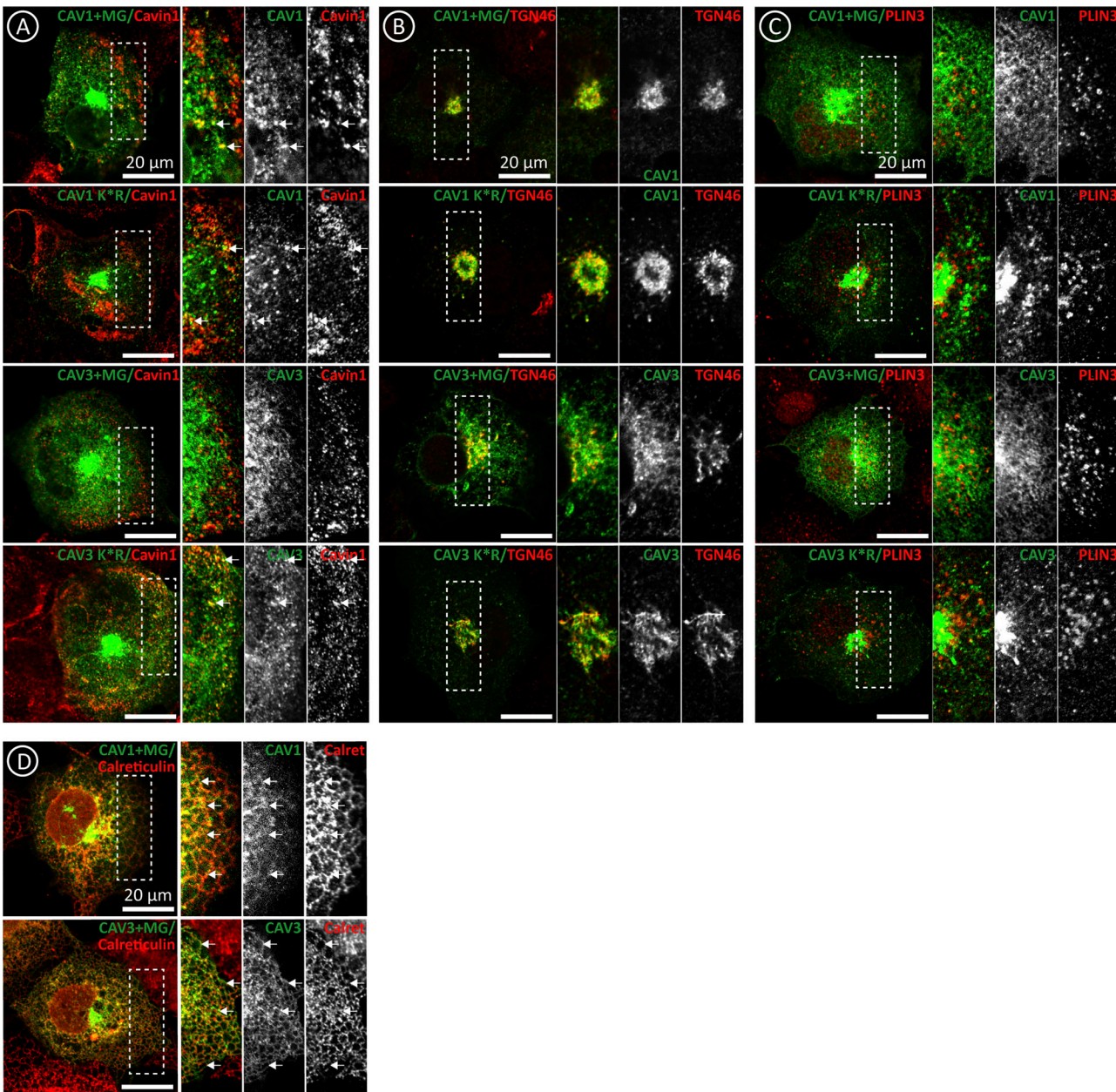

Figure S3.    **Proteasomal degradation determines trafficking of newly synthesized caveolins.** Supplemental for Fig. 3. **(A–D)** CAV1 and CAV3 (in CTL or in cells additionally treated with MG132) and CAV1 K*R and CAV3 K*R were expressed for 3 h. Proteins were colocalized with antibodies recognizing Cavin-1 (PM caveolae), TGN46 (GC), Calreticulin (ER), and Plin-3 (LDs) and analyzed by confocal microscopy. Scale bars are 20 µm.

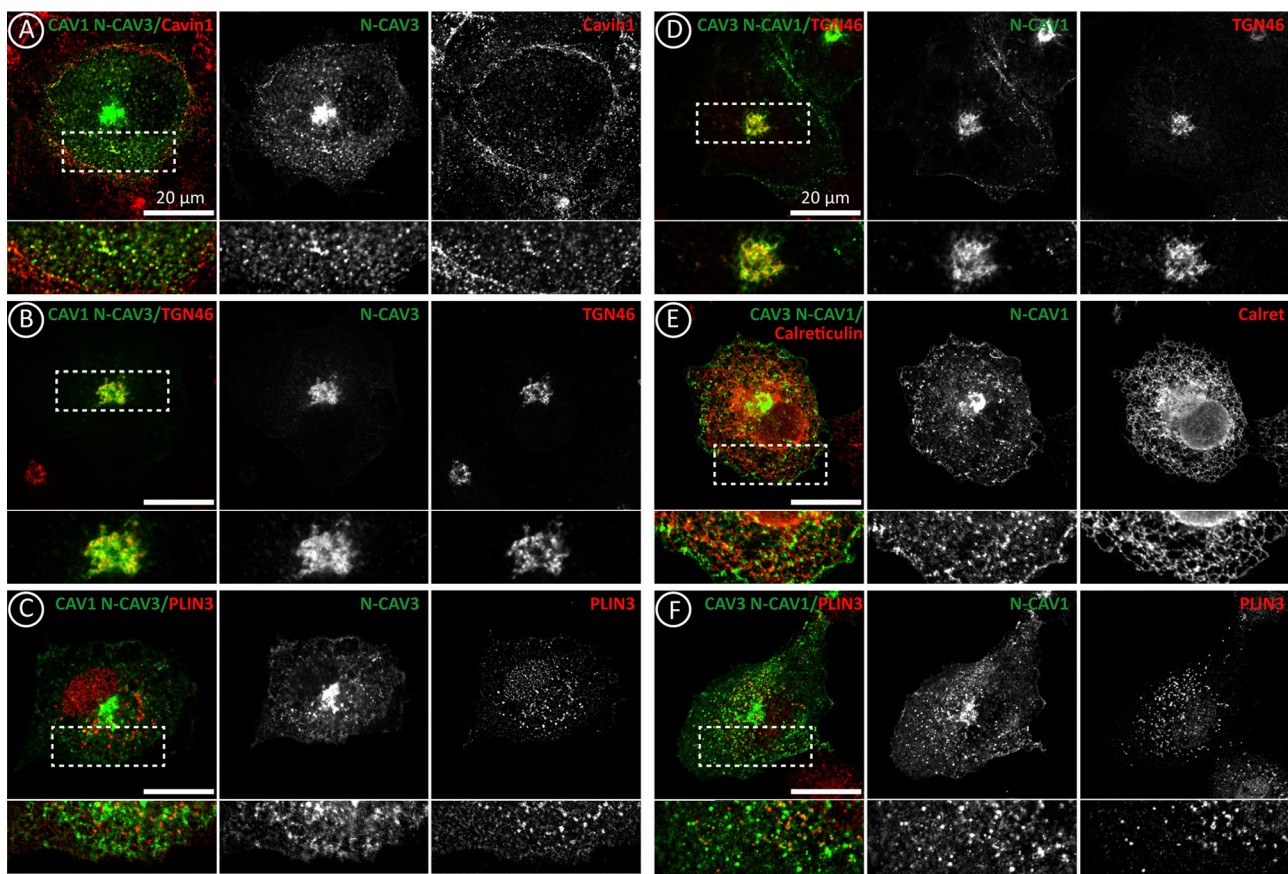

Figure S4. **Caveolin N-terminus determines the localization of newly synthesized caveolins.** Supplemental for Fig. 4. **(A–F)** CAV1 N-CAV3 and CAV3 N-CAV1 were expressed for 3 h. Proteins were colocalized with antibodies recognizing Cavin-1 (PM caveolae), TGN46 (GC), Calreticulin (ER), and Plin-3 (LDs) and analyzed by confocal microscopy. Scale bars are 20 µm.

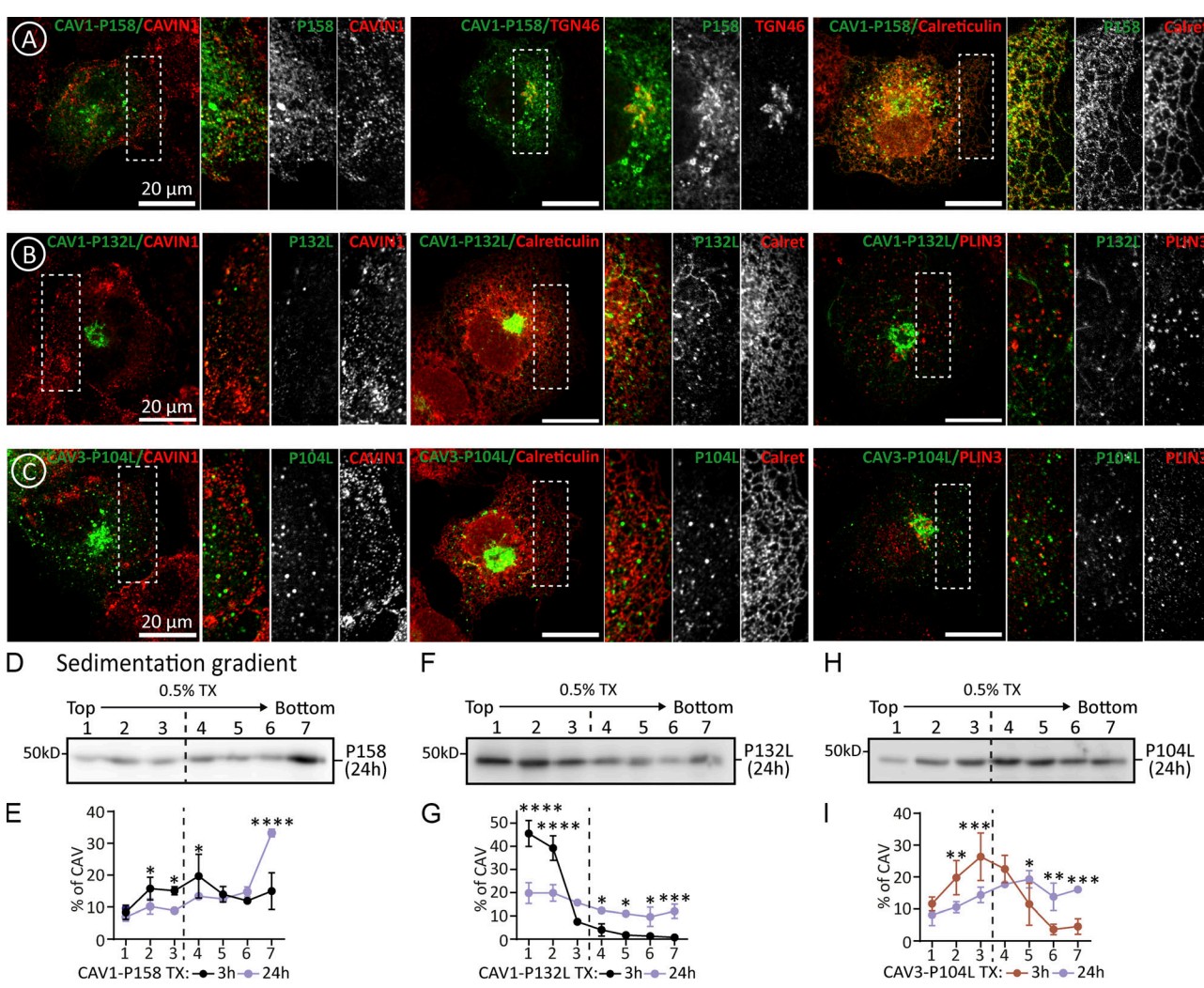

Figure S5. **Intracellular distribution of caveolin pathogenic mutants.** Supplemental for Figs. 5 and 6. **(A–C)** CAV1-P158, CAV1-P132L, and CAV3-P104L were expressed for 3 h. Proteins were colocalized with antibodies recognizing Cavin-1 (PM caveolae), TGN46 (GC), Calreticulin (ER), and Plin-3 (LDs) and analyzed by confocal microscopy. **(D–I)** Mutants were expressed for 24 h. Cells were solubilized in TX and fractionated in sedimentation gradients as in Fig. 2 K. The IB shows representative experiments and the graphs the average distribution when compared to the 3 h expression ($n \geq 3$ independent experiments). All graphs show means ± SD; ns, not significant; *$P < 0.05$, **$P < 0.01$, ***$P < 0.001$, ****$P < 0.0001$ in a two-way ANOVA test (E, G, and I). Scale bars are 20 µm. Source data are available for this figure: SourceData FS5.

**Provided online is Table S1. Table S1 shows the key resources used in this study.**

