## [Peer Review File · The Journal of Cell Biology]

Early proteostasis of caveolins synchronizes trafficking, degradation, and oligomerization to prevent toxic aggregation

Frederic Morales-Paytuví, Alba Fajardo, Carles Ruiz-Mirapeix, James Rae, Francesc Tebar, Marta Bosch, Carlos Enrich, Brett Collins, Robert Parton, and Albert Pol

Corresponding Author(s): Albert Pol, Consorci Institut D'Investigacions Biomediques August Pi I Sunyer and Robert Parton, University of Queensland

Review Timeline:

Submission Date:	2022-04-07
Editorial Decision:	2022-05-16
Revision Received:	2023-04-05
Editorial Decision:	2023-05-18
Revision Received:	2023-06-03
Accepted:	2023-06-09

Monitoring Editor: Elizabeth Miller

Scientific Editor: Andrea Marat

Transaction Report:

DOI: <https://doi.org/10.1083/jcb.202204020>

May 16, 2022

Re: JCB manuscript #202204020

Dr. Albert Pol
Institut d'Investigacions Biomèdiques August Pi i Sunyer (IDIBAPS)
Lipid Trafficking and Disease Group
Centre de Recerca Biomèdica CELLEX (IDIBAPS), Office B12
Casanova 143, 08036
Barcelona 08036
Spain

Dear Dr. Pol,

Thank you for submitting your manuscript entitled "Proteostatic regulation of caveolins avoids premature oligomerisation and preserves ER homeostasis". The manuscript has been evaluated by expert reviewers, whose reports are appended below. Unfortunately, after an assessment of the reviewer feedback, our editorial decision is against publication in JCB.

The reviewers and editors all have enthusiasm for the topic, the broad findings, and the timing for the study, but we all share several concerns that would need to be addressed before we consider the study suitable for JCB. We suggest that additional experiments should focus on improved co-localization (as suggested by both reviewers), and on better analysis of oligomerization state. An underlying editorial concern was the question of cause-and-effect among the various treatments/mutants: lipid binding, ER retention, degradation and trafficking are all inter-related, but dissecting the dominant effects would go a long way towards clarifying mechanism. The mutants you have in hand might be a good way to dissect this, as suggested by Reviewer 2.

Therefore, although your manuscript is intriguing, I feel that the points raised by the reviewers are more substantial than can be addressed in a typical revision period. If you wish to expedite publication of the current data, it may be best to pursue publication at another journal.

Given interest in the topic, I would be open to resubmission to JCB of a significantly revised and extended manuscript that fully addresses the reviewers' concerns and is subject to further peer-review. If you would like to resubmit this work to JCB, please contact the journal office to discuss an appeal of this decision or you may submit an appeal directly through our manuscript submission system. To appeal the decision please provide a detailed point-by-point response and plan outlining how you propose to address the reviewer concerns, so that we may provide feedback and potentially discuss with the reviewers to ensure everyone is on the same page as to what is expected for resubmission to JCB. Please note that priority and novelty would be reassessed at resubmission.

Regardless of how you choose to proceed, we hope that the comments below will prove constructive as your work progresses. We would be happy to discuss the reviewer comments further once you've had a chance to consider the points raised in this letter. You can contact the journal office with any questions, cellbio@rockefeller.edu or call (212) 327-8588.

Thank you for thinking of JCB as an appropriate place to publish your work.

Sincerely,

Elizabeth Miller, PhD
Monitoring Editor

Andrea L. Marat, PhD
Senior Scientific Editor

Journal of Cell Biology

Reviewer #1 (Comments to the Authors (Required)):

This work by Morales-Paytuví et. al., addresses how degradation regulates the oligomeric status and subsequent trafficking of caveolin homologues. The work proposes that, in contrast to the current view, caveolins oligomers (8S complex) should form in the Golgi instead of ER and premature oligomerization is prevented by caveolin degradation by ERAD/proteasome. The authors

use transient transfection of caveolins in COS cells together with suppression of protein translation using cycloheximide as model system. Hereby, they study protein localisation, oligomerisation, trafficking and degradation following translation as detected by fluorescent microscopy and immunoblotting. Authors claim that caveolin homologues form similar types of oligomers but that, expression levels, degradation, trafficking and oligomeric status varies. An N-terminal ER-retention signal is identified in Cav3 and cholesterol and fatty acids influence trafficking. Cav1 and Cav3 mutants which cannot be degraded by the proteasome induce ER stress and influence oligomerisation and trafficking similarly to mutants which influence oligomerisation. The presented hypothesis, that oligomerisation of caveolins influence the exposure of sorting signals for trafficking, is based on the cryoEM structure of Cav1 oligomers (Porta et al., 2022 BioRxiv) and previous studies showing that antibodies that only recognize oligomeric or monomeric caveolin. Figures are clear and the text is easy to follow, although the reasoning and logic of the results is sometimes hard to follow whereby there is a risk for misunderstandings or overinterpretations of the data. The hypothesis is relevant and interesting, but unfortunately many of the individual observations and conclusions have not been thoroughly followed up (see specific points below) and some results and conclusions might be found as circumstantial. Some statements are bold without direct strong supporting data. Therefore, the work would benefit from focusing on some key findings and investigate these thoroughly to provide string conclusions.

Specific points raised:

1. Figure 1 presents the structure of Cav1 oligomers based on (Porta et al., 2022 BioRxiv) and Alfold models showing that Cav3 likely form similar types of oligomers. In Fig 1B the legend states that "the structure was manually modelled" but I could not find information of how this was done? Is this MD simulations of the position of caveolin in the membrane, or just modelled? Do we learn anything more from this image as compared to (Porta et al., 2022 BioRxiv). If not, I propose that Fig1A should be limited to only include relevant data from (Porta et al., 2022 BioRxiv) that cant be referenced, so that it is clearer that this work include the primary data. The following subfigures of Fig. 1 would then more clearly show the comparison done in this work in terms of especially Cav3.
2. Figure 2 is comparing the differences localization and complex formation between caveolins. In Fig. 2C, representative fluorescent images are shown of the localization of the caveolins. The different organelles are indicated with arrows. Yet, as I understand there are no markers of the respective organelles. Since this is quantified, and major conclusions are being drawn from the different temporal localizations in the manuscript, the localization would need to be verified. Furthermore, in the left panel with Cav1 the empty arrow head is pointing to a line of stain. Yet, if Cav1 is located to caveolae in the plasma membrane most of the stain would be in puncta (which also is visible in this image) and the amount will depend on the focal plane of the microscopy. Yet, punctate stain could also be derived from vesicles and endosomal like structures. It is difficult to judge what is really quantified as plasma membrane and other organelles in these images. Same is valid for data in 3C-D, 4B and E and 5 B,C,D,E etc
3. How did the quantification (discussed in point 2) account for the 3D localization in cells with stain in different focal planes? Image analysis and quantification is not described.
4. As I understand, Cav3 is expressed mainly in muscle cells where it localizes to caveolae in the plasma membrane. Yet, in this work Cav3 is mainly localized to lipid droplets and rarely at the plasma membrane. What is the reason for this? and could other complementary model systems be used to address this the proteostatic regulation of Cav3 in the presence of relevant tissue specific components? It would strengthen the work if this was followed up on.
5. The N-term of Cav3 is proposed to harbor an ER retention signal. It is however not clear why Cav3 would need such signal and not Cav1 if both should be trafficked to the plasma membrane? Also, it is not clear how and when the N-terminus acts as a retention signal which could explain the reason for this. Could be followed up.
6. Based on the data in Fig 3A, the authors state that newly synthesized caveolins have a short life time. In order to conclude this the degradation of the endogenous proteins has to be addressed where the proteins are not overexpressed by transient expression.
7. The work also addresses the impact of lipids on proteasomal degradation. Yet, I cant find information of how the how added and how much? Is the interpretation that if cholesterol and fatty acids are present, protein folding and oligomerization is improved and therefore there is less degradation? Adding fatty acids would increase lipid droplet formation and therefore Cav3 can localize there. This could be followed up on.
8. I am not certain if the authors are proposing that ERAD prevents oligomerization by reducing levels of expression or recognizing oligomers or missfolded protein. It is stated that "we hypothesized that this system has evolved to maintain low levels of caveolins within the ER and minimise oligomerization". Maybe this could be made clearer throughout the manuscript?
9. Title says "preserves ER homeostasis" After reading the work I am not sure what this refers to. Is it that if proteins are overexpressed, degradation preserves ER homeostasis and prevent UPR?

Reviewer #2 (Comments to the Authors (Required)):

Morales-Paytavi et al. characterize the early steps of caveolins (CAV) 1 and 3 assembly, oligomerization and traffic within the secretory pathway. The authors initially model CAV1, 2 and 3 using alphaFold and CAV1 structure as a reference. This modelling supports previous findings by the authors and others explaining phenomena such as preferential binding regions for antibodies. The authors continue to characterize CAV 1 and 3 looking at their cellular localization and oligomeric state making note that CAV3 is found primarily at low abundance and as monomers or low oligomeric states and on LDs and the ER. This is in contrast to CAV1 found in the Golgi and plasma membrane and exhibits two high oligomeric species. The authors further detail the proteosomal degradation of caveolins showing regulation by neutral lipid abundance, preferential degradation of monomeric caveolins, and by using degradation insensitive mutants provide evidence that ERAD prevents the formation of CA oligomers in the ER. Lastly, the authors explore the determinants for CAV3 retention in the ER and how characterized pathogenic point mutations, can lead to accumulation of high order structures thus providing a mechanistic explanation to caveolin dependent pathogenic conditions.

The basic aspects of caveolin oligomerization and trafficking is fascinating and deserves further characterization. In this respect, this study is timely and focus on an important problem in cell biology. The distinct behaviour of CAV1 and CAV3 is very interesting and convincing. On the other hand, the importance of oligomerization and competing degradation on the trafficking and distribution of Caveolins appears premature and in my opinion requires additional work. A few specific comments are below.

Major comments:

1) All the microscopy to assess the localization of Caveolins should include organellar markers. Without organelle markers it is unclear how the authors assign punctate structures to ER or GC, which is critical for most of the conclusions taken by the authors. The use of markers is a standard in the cell biology field and should be used.

2) The authors observe clear differences in the localization and behavior on the gradient of CAV1 and CAV3. However, the conclusions on their distinct oligomeric states are premature. The oligomeric state of caveolins is analysed by density gradients after mild solubilisation with 0,5% Triton X-100. Is it clear that under these mild conditions caveolins are fully solubilized? The presence of any lipids or protein bound to caveolins under these conditions will affect their migration and compromise the conclusion regarding their oligomeric state. For example, in Figure 2F it is shown that CAV3 floats to the top of the gradients. Rather than the monomeric/low oligomeric state, this migration pattern probably reflects CAV3 association with the light lipid droplet fraction. In this reviewer's opinion further validation of the assay is required to demonstrate its suitability in determining caveolins oligomeric states, a central point of the manuscript.

3) There are convincing differences in the steady levels of Caveolins in cells treated with proteasome inhibitors or expressing the K*R mutants. However, the connections between oligomerization and ERAD/proteasomal degradation are not obvious. First, the effect of proteasome inhibition in caveolin oligomerization is not directly tested. Second, the results with the K*R mutants is confusing. It is stated that "These mutants retain the capacity to traffic to the PM". However, this is not supported by the data presented in Figure 3, showing that both CAV1 and CAV3 K*R mutations localize to ER and GC. CAV1 K*R migrates mainly to the bottom of the gradient. It is not clear that this is an oligomer or an aggregate. Third, the lipid regulation experiments could be potentially interesting but no treatment controls should have been included. In addition, control proteins that do not respond to the lipid treatments should have been included in the analysis.

4) The phenotypes of the pathogenic mutations are very subtle mild. It is difficult to appreciate the difference between Fig5C and CAV3 in Fig3D. Also the authors make a point that ERAD can clear low oligomeric/monomeric CAV. I would therefore suspect that the pathogenic mutants should be easily cleared since they cannot oligomerize. However, their levels at the top of the velocity gradient seem very abundant and only slightly oligomerize during lengthy expression. It would be important to discuss this discrepancy as to why these mutants are not cleared despite being at a low monomeric conditions. What happens at further time points of expression?

Minor comments:

1) Introduction, paragraph 2: "...examine the early proteostasis of caveolin family members and mutants". Proteostasis is jargon with very loose meaning. In this context and at other places of the manuscript I would suggest authors to be more specific about the question they aim to address.

2) Caveolins do not possess a signal peptide or a transmembrane domain. Is it known how they are targeted to the ER? It would be interesting to give additional background information about this.

3) The degradation of Caveolins by ERAD is mentioned multiple times. However, only inhibition of the proteasome, which degrades many substrates, was tested. Therefore, the authors should consider to change their terminology.

4) In Figure 3D it would be important to show that CAV3 expressing cells have LDs.

5) In Figure 4 (and others), protein steady state levels are analyzed. This is different from protein stability and the two terms should not be used interchangeably.

6) The authors should provide more details on how the expression of XBP1S and CHOP was determined.

7) It would be interesting to discuss what the roles for CAV3 retention in the ER and on LDs might be in a physiological context.

8) Based on the modelling, could the authors speculate how CAV2 would behave if tested in this study.

Barcelona, April 5th, 2023

Re: JCB manuscript #202204020 by Morales-Paytuví et al.

Dear Dr. Marat,

We are delighted by the shared enthusiasm of the Editors and Reviewers for the topic, findings, and timing of the manuscript. We greatly appreciate all the suggestions and advice that undoubtedly have improved the flow, significance, and impact of our work. We appreciate the opportunity to submit this fully revised manuscript which follows the proposed revision plan approved in June 2022.

The initial contact with experts has been extremely helpful for us to clarify the most relevant data, and the required new terminology to present these results, and to design new assays to test specific aspects of the model (including assays to distinguish functional oligomers from aggregates). Comparing a plethora of proteostatic mechanisms functioning very early in the biosynthesis of eleven different caveolins and four different organelles has been conceptually and methodologically challenging but we believe that the new manuscript proposes a convincing unifying model of the early proteostasis of caveolin family members and paradigmatic mutants.

The new model integrates many previous unconnected and puzzling observations in the field with the recently published caveolin oligomeric structure and, in addition, illuminates novel mechanistic insights of the pathogenic potential of caveolins leading to disease. We have summarized all these implications in a new Discussion.

As detailed in the point-by-point letter, the main additions to the new version are:

- i) All caveolins and mutants have now been analyzed using an independent biochemical assay to distinguish between functional caveolin oligomers and non-functional aggregates.
- ii) We have analyzed the frameshift CAV1 mutation associated with pulmonary arterial hypertension, CAV1 P158PfsX22, that is unable to reach late-Golgi and thus, unable to oligomerize, rapidly forming non-functional aggregates, and triggering ER stress.
- iii) We have used correlative light and electron microscopy (CLEM) to visualize the effect of degradation-resistant mutant caveolins on cellular ultrastructure, revealing a dramatic effect on Golgi morphology.

- iv) A more concise title: "*Dynamic early proteostasis of caveolins balances trafficking and degradation to reduce cytotoxicity*".
- v) We have included additional explanations throughout the text to clarify conclusions and importantly, to address the cause-and-effect question raised by the Editors. A new Discussion has been written. Furthermore, we have prepared a final scheme (Fig. 8) and a graphical abstract to help with the interpretation.
- vi) We have quantified the colocalization of all caveolins and mutants with the involved organelles (ER, GC, LD, and PM) in confocal microscopy images.

Please note that due to the volume of additional experiments the current manuscript is in the format of an "Article" rather than a "Report" and that now includes an additional author.

Once again, we would like to thank you for your continued support. We hope that the manuscript is now suitable for publication in *The Journal of Cell Biology* and look forward to hearing from you in due course.

Albert Pol and (p.p.) Rob Parton.

Point-by-point

Editorial Comments

The reviewers and editors all have enthusiasm for the topic, the broad findings, and the timing for the study, but we all share several concerns that would need to be addressed before we consider the study suitable for JCB.

We are delighted with the comments about the quality and timing of our work.

We suggest that additional experiments should focus on improved co-localization (as suggested by both reviewers),

The eleven characterized caveolins have been co-localized with organellar markers in confocal microscopy images. The Manders overlap coefficient (M2) has been used to quantify the co-localization and the resulting graph is now in each figure. Images are included mainly as supplemental figures and only the most relevant co-localizations are in the main figures. These more in-depth studies have confirmed the differential proteostasis of caveolin family members and mutants proposed in the original manuscript using conventional microscopy and manually annotated.

and on better analysis of oligomerization state.

To determine if caveolins are forming oligomers or non-functional high-molecular-weight aggregates, like those formed by mutants failing to oligomerize (details in the new Figs. 5 and 6), we have applied an additional biochemical criterion. The nature of the complexes has been evaluated after solubilization of cells with a combination of 0.2% TX and 0.4% SDS. Upon these conditions, the 8S oligomers remain as insoluble complexes but aggregates are largely solubilized into monomeric caveolins (Copeland et al., 2017; Hayer et al., 2010a).

An underlying editorial concern was the question of cause-and-effect among the various treatments/mutants: lipid binding, ER retention, degradation and trafficking are all inter-related, but dissecting the dominant effects would go a long way towards clarifying mechanism. The mutants you have in hand might be a good way to dissect this, as suggested by Reviewer 2.

We sincerely hope that these concerns have been resolved with the new mutants, biochemical approaches, and microscopy studies. We think that the concerns of Reviewer 2 have been completely addressed.

Given interest in the topic, I would be open to resubmission to JCB of a significantly revised and extended manuscript that fully addresses the reviewers' concerns and is subject to further peer-review.

Once again, we appreciate this opportunity. The manuscript has been extended into a regular "Article".

Response to general comments:

As outlined in our previous letter to the editor, we appreciate the suggestion of Reviewer 1 that some results could be followed up and we firmly believe that our study will open new lines of investigation. The important findings presented here include identification of the sites of caveolin oligomer formation; elucidation of the crucial role of ER-mediate proteasomal degradation in reducing caveolin protein toxicity and allowing its maturation; identification of differences in caveolin isoform trafficking and the regulation by specific lipid species; and the effect of disease mutations in this new scheme.

We would like to emphasize that our new findings are now consistent with numerous puzzling observations in previous studies, as well as with the newly described structure of the caveolin

oligomer, allowing us to generate what we believe will form a new consensus view of the caveolin pathway. To better illustrate the major findings of the study, we now present a scheme (Fig. 8) which will provide a new universal working model for caveolin trafficking.

Reviewer 1 raises a few additional interesting questions (detailed in the point-by-point rebuttal) that we believe are outside the scope of this study as discussed in our previous letter to the editor. However, we have included additional text to acknowledge that further studies are needed from here for understanding the mechanistic details and biological implications of some novel phenotypes. Furthermore, we suggest reasonable hypotheses for assisting future research. We hope that the new implications for caveolin-related diseases, which are now clearly outlined in the revised manuscript, demonstrate the importance of the current study and the general interest in the new findings.

Reviewer 1 also suggested the analysis of endogenous caveolins using our experimental systems. We have performed the suggested experiments (point 6 in the point-by-point rebuttal) and confirm that, like the transfected proteins, endogenous caveolins tend to form aggregates and display trafficking defects when the proteasome is inhibited. Although in our opinion the use of drugs has inevitable limitations, if the editors deem it appropriate, we will include these results as a supplementary figure.

Finally, we leave it to the discretion of Editors whether the use of the word “proteostasis” is appropriate. We were also initially hesitant about using the term “proteostasis” but after reading recent articles (including (Hipp et al., 2019) *Nat Rev Mol Cell Biol*) which revealed a potential link between these processes and the poorly understood and unresolved role of caveolins in long-term pathogenic conditions, we believe the term is useful for the field and appropriate here. If the editors think that the term is “jargon”, we are open to other suggestions.

Reviewer #1:

... Figures are clear and the text is easy to follow, although the reasoning and logic of the results is sometimes hard to follow whereby there is a risk for misunderstandings or overinterpretations of the data. The hypothesis is relevant and interesting,

We thank the Reviewer for the critical reading of the manuscript and suggestions and for the comment about relevance and interest of the hypothesis. We completely agree that the previous version was lacking key experiments and explanations. As detailed in the cover letter, we have included new experimental data and additional text that, in our opinion, now fully support the hypothesis.

but unfortunately many of the individual observations and conclusions have not been thoroughly followed up (see specific points below) and some results and conclusions might be found as circumstantial. Some statements are bold without direct strong supporting data. Therefore, the work would benefit from focusing on some key findings and investigate these thoroughly to provide string conclusions.

As detailed in the cover letter, new supporting data that includes other mutants, new biochemical analyses, additional quantitation, and new confocal and electron microscopy analyses have been incorporated into this version. We hope that this reformatted version supports the hypothesis by providing a convincing unifying model explaining the early proteostasis of caveolins.

Specific points raised:

1. Figure 1 presents the structure of Cav1 oligomers based on (Porta et al., 2022 BioRxiv) and AlphaFold models showing that Cav3 likely form similar types of oligomers. In Fig 1B the legend states that "the structure was manually modelled" but I could not find information of how this was

done? Is this MD simulations of the position of caveolin in the membrane, or just modelled? Do we learn anything more from this image as compared to (Porta et al., 2022 BioRxiv). If not, I propose that Fig1A should be limited to only include relevant data from (Porta et al., 2022 BioRxiv) that cant be referenced, so that it is clearer that this work include the primary data. The following subfigures of Fig. 1 would then more clearly show the comparison done in this work in terms of especially Cav3.

We apologize for not properly explaining the initial figure. Figure 1B (now Figure 1D) is not the result of an MD simulation. It is simply the cryoEM structure of CAV1 placed within a lipid bilayer using PyMOL to provide a visual scale for the proposed model of how CAV1 inserts into the inner membrane leaflet. We have amended this text to make it easier to understand. Although a cartoon model of CAV1 in the membrane was shown in Porta et al. (Porta et al., 2022), it was not drawn to scale. We believe including this updated model in our manuscript provides greater clarity for the reader, by giving direct context regarding the proposed membrane interactions of caveolin-oligomers at the plasma membrane.

2. Figure 2 is comparing the differences localization and complex formation between caveolins. In Fig. 2C, representative fluorescent images are shown of the localization of the caveolins. The different organelles are indicated with arrows. Yet, as I understand there are no markers of the respective organelles. Since this is quantified, and major conclusions are being drawn from the different temporal localizations in the manuscript, the localization would need to be verified. Furthermore, in the left panel with Cav1 the empty arrow head is pointing to a line of stain. Yet, if Cav1 is located to caveolae in the plasma membrane most of the stain would be in puncta (which also is visible in this image) and the amount will depend on the focal plane of the microscopy. Yet, punctate stain could also be derived from vesicles and endosomal like structures. It is difficult to judge what is really quantified as plasma membrane and other organelles in these images. Same is valid for data in 3C-D, 4B and E and 5 B,C,D,E etc

In the original version we tried to minimize the number of figures for a “Report format”. We understand now that a better characterization of the protein distribution was essential. Thus, the different caveolin proteins characterized in our study have now been co-localized with organellar markers in confocal microscopy images. The Manders overlap coefficient (M2) has been used to quantify the co-localization and the resulting graph is now in each figure. Images are included mainly as supplemental figures and only the most relevant co-localizations are in the main figures. These more in-depth analyses have confirmed the differential proteostasis of caveolins and mutants proposed in the original manuscript using conventional microscopy and manually annotated. Details are included in the Materials and Methods section.

3. How did the quantification (discussed in point 2) account for the 3D localization in cells with stain in different focal planes? Image analysis and quantification is not described.

The analysis was based on conventional fluorescence microscopy to allow simultaneous visualization of the whole cell, analysis required to compare the relative presence of new caveolins between four different organelles. We now understand that additional details were required for readers not familiarized with the complex intracellular distribution of caveolins (see point 2). Details of the analysis are now included in the Material and Methods section.

4. As I understand, Cav3 is expressed mainly in muscle cells where it localizes to caveolae in the plasma membrane. Yet, in this work Cav3 is mainly localized to lipid droplets and rarely at the plasma membrane. What is the reason for this?

We apologize for not explaining this point properly. Indeed, CAV3 is a PM resident protein with the capacity of forming caveolae not only in muscle cells but in all the cell types tested so far. When CAV1 or CAV3 are expressed for 24 hours both proteins are largely indistinguishable and

accumulate on the PM, as repeatedly demonstrated in our previous works (Luetterforst et al., 1999; Pol et al., 2001). However, our new model cell system of synchronized expression followed by analyses at very early time points has detected that this trafficking is much slower for CAV3 than for CAV1. Because the half-life of caveolins on the PM has been estimated in more than 24 hours but intracellular caveolins are rapidly transported or degraded, this difference was unnoticed until now. This explanation is included in the new version (page 6).

and could other complementary model systems be used to address this the proteostatic regulation of Cav3 in the presence of relevant tissue specific components? It would strengthen the work if this was followed up on.

Considering this possibility, in the initial version of the manuscript we analyzed the distribution of newly-synthesized caveolins in C2C12 myoblasts (Fig. S2). The differential trafficking of CAV1 and CAV3 was even more striking in this model muscle cell line. This point is now explained more clearly in the text (page 6).

5. The N-term of Cav3 is proposed to harbor an ER retention signal. It is however not clear why Cav3 would need such signal and not Cav1 if both should be trafficked to the plasma membrane?

We do agree with the Reviewer that this is an intriguing trait of CAV3 that, predicted *in silico* by AlphaFold, we have initially tested here. We have included additional explanations in the new Discussion to make clear that the physiological relevance of this phenotype needs further analyses (page 12).

Unfortunately, the role of caveolins in the ER/LDs encloses an unexpected complexity that we and others have not been able to clarify since our original description in *JCB* 20 year ago (Pol et al., 2001), even though the presence of caveolins on LDs has been repeatedly confirmed by other groups in cell biology, metabolism, and proteomic studies (Parton et al., 2020; Pol et al., 2020). Hence, showing the different early proteostasis of CAV1 and CAV3 and the higher affinity of CAV3 for LDs, although initial indeed, is an interesting starting point that will help future research in the caveolin and the LD fields (page 18).

Also, it is not clear how and when the N-terminus acts as a retention signal which could explain the reason for this. Could be followed up.

This is also a complex question. Our previous analysis demonstrated that this domain is not essential for targeting CAV3 to the PM (Luetterforst et al., 1999) or into LDs (Pol et al., 2001) and its role has been traditionally a mystery. It was suggested that it could be part of a pH-sensitive helix region (Kim et al., 2016) but its physiological relevance is unknown (Park et al., 2021).

The initial residues of CAV3 are quite unique but highly conserved among CAV3 of different species. We have been unable to identify similar sequences in other proteins and only detected a relative similarity with few enzymes (methyltransferase, kinases, ATPases, etc) of prokaryotes and lower eukaryotes (with in principle no caveolin homologs). Further, CAV3 is unique among all these proteins because it is the only one with the domain in the N-terminal side. Thus, these amino acids do not correspond to an established functional motif that we can test straightforward.

Although the precise mechanism of action is unclear, and even though it is an initial observation, in the context of our study the fact that CAV1 is efficiently retained in the ER when these amino acids are exchanged, is needed to prove two key points of the hypothesis: i) that CAV3 is retained for longer times in the ER and ii) that ER retention increases CAV degradation. We recognize throughout the text that additional studies are needed for understanding the mechanism of action of the motif (page 12).

6. Based on the data in Fig 3A, the authors state that newly synthesized caveolins have a short life time. In order to conclude this the degradation of the endogenous proteins has to be addressed where the proteins are not overexpressed by transient expression.

Following the Reviewer's suggestion, we have applied our experimental systems to study endogenous CAV1. A slight but significant increment (14%) in the cellular levels of CAV1 were quantified when the proteasome was inhibited with MG132 for three hours (A), which is consistent with previous experiments using a metabolic labelling approach (Hayer et al., 2010b).

CAV1 distribution was analyzed by fluorescence microscopy using a monoclonal antibody that only recognizes non-oligomeric newly synthesized CAV1 (Pol et al., 2001; Pol et al., 2004; Pol et al., 2005). As expected, in control cells, this CAV1 pool is in the GC (B left). When these cells were treated with MG132 for three hours, the GC pool of CAV1 was drastically reduced (B center), likely reflecting the trafficking defects observed with transfected caveolins. After 16 hours with the drug, only punctate structures were recognized by the antibodies in some cells (B right).

When COS cells were analyzed with polyclonal antibodies (recognizing all the caveolin pools), PM membrane CAV1 was now evident, and the GC pool commonly observed (C). When these cells were treated for 16 hours with MG132, the PM pool of the protein was drastically reduced (D) with CAV1 retained intracellularly. In sedimentation gradients, we did not observe changes in endogenous caveolins in cells treated for three hours with MG132 (data not shown). However, when cells were treated with MG132 for 16 hours, CAV1 tended to equilibrate into higher-molecular-weight species that were efficiently solubilized by TX+SDS, as expected for aggregates (E and F).

These results suggest that, like the transfected proteins, endogenous caveolins display trafficking defects when the proteasome is inhibited. Endogenous caveolins also tend to form aggregates, as observed after 16 hours of the treatment with MG132.

However, after 16 hours with the proteasomal inhibitor the ER collapsed, as visualized with a transfected GFP-TANGO (a marker of ER exit sites) (G). In these cells, CAV1 was visible accumulated in punctate structures along the collapsed ER (G). Hence, in this case, the impaired trafficking of caveolins to the PM could be the result of general defects of the exocytic pathway rather than reflecting that impaired degradation of caveolins promotes aggregation and leads to trafficking defects. Therefore, although these long treatments are commonly used for studying the stability of particular proteins, we cannot ensure that these results are strictly reflecting a specific trait of caveolins. This is the main reason why in this work we have prioritized the use of non-degradable mutants rather than chemical inhibitors affecting other proteins. In any case, these results confirm that when intracellularly retained by environmental causes, caveolins tend to aggregate.

We leave it to the discretion of Editors whether the experiment suggested by the Reviewer could be added, for example, as a supplementary figure.

7. The work also addresses the impact of lipids on proteasomal degradation. Yet, I cant find information of how the how added and how much?

We apologize for the lack of detail. Treatments and doses are now included.

Is the interpretation that if cholesterol and fatty acids are present, protein folding and oligomerization is improved and therefore there is less degradation?

Although described by us and others (Hayer et al., 2010a) the precise role of cholesterol during the trafficking of CAV1 is still unclear. CAV1 is a cholesterol binding protein (Hulce et al., 2013) and cells without CAV1 accumulate cholesterol in the ER and Golgi (Bosch et al., 2011a; Bosch et al., 2011b), suggesting a direct role of cholesterol in CAV1 trafficking within the ER and during the rest of the biosynthetic pathway. Whether CAV3 binds cholesterol is unknown, although our unpublished data suggest that CAV3 has a much lower affinity for cholesterol than CAV1. Additional explanations have been included in the Discussion (page 17).

Adding fatty acids would increase lipid droplet formation and therefore Cav3 can localize there. This could be followed up on.

This process has been mechanistically characterized in lower eukaryotes (Ruggiano et al., 2016). By increasing the number of LDs, cells increase the stability of LD-resident proteins. To illustrate that this is a general mechanism, we have characterized ALDI that, like CAV3, is more stable in cells loaded with LDs (Fig. 3). Additional explanations have been included in the Discussion (page 9 and 17).

8. I am not certain if the authors are proposing that ERAD prevents oligomerization by reducing levels of expression or recognizing oligomeres or missfolded protein. It is stated that "we hypothesized that this system has evolved to maintain low levels of caveolins within the ER and minimise oligomerization". Maybe this could be made clearer throughout the manuscript?

The Reviewer raises a crucial point that needed to be corrected. The new biochemical criterion allows us to differentiate between functional oligomers and non-functional caveolin aggregates. The new version clearly differentiates between functional oligomers (forming only in late-Golgi) and non-functional aggregates (forming along the biosynthetic pathway). We have included new experimental data, biochemical analysis, and additional details to clarify this key point.

To illustrate that caveolin degradation avoids aggregation, we have analyzed the frameshift CAV1 mutation associated with pulmonary arterial hypertension. CAV1 P158PfsX22 is unable to reach late-Golgi and thus, unable to oligomerize. When the proteasome is inhibited, P158 rapidly forms non-functional aggregates and triggers ER stress (Fig. 5). Additional explanations have been included throughout the manuscript and a final model have been prepared (Fig. 8).

9. Title says "preserves ER homeostasis" After reading the work I am not sure what this refers to. Is it that if proteins are overexpressed, degradation preserves ER homeostasis and prevent UPR? A new title is proposed.

Reviewer #2:

The basic aspects of caveolin oligomerization and trafficking is fascinating and deserves further characterization. In this respect, this study is timely and focus on an important problem in cell biology. The distinct behaviour of CAV1 and CAV3 is very interesting and convincing.

We thank the Reviewer for the critical reading of the manuscript, excellent comments, and useful suggestions.

On the other hand, the importance of oligomerization and competing degradation on the trafficking and distribution of Caveolins appears premature and in my opinion requires additional work.

We completely agree that the previous version was lacking key experiments and explanations. We hope that the new experimental data and explanations reasonably demonstrate now the proposed scheme of the early proteostasis of caveolins.

Major comments:

1) All the microscopy to assess the localization of Caveolins should include organellar markers. Without organelle markers it is unclear how the authors assign punctate structures to ER or GC, which is critical for to most of the conclusions taken by the authors. The use of markers is a standard in the cell biology field and should be used.

In the original version we tried to minimize the number of figures for a "Report format". We understand that a better characterization of the protein distribution was essential. Hence, the eleven caveolins characterized in our study have been co-localized with organelle's markers in confocal

microscopy images. The Manders overlap coefficient (M2) has been used to quantify the co-localization and the resulting graph has been incorporated in each figure. Images are included mainly as supplemental figures and only the most relevant co-localizations are in the main figures. These more in-depth analyses have confirmed the differential proteostasis of caveolins and mutants proposed in the original manuscript using conventional microscopy and manually annotated. Details are included in the Materials and Methods section.

2) The authors observe clear differences in the localization and behavior on the gradient of CAV1 and CAV3. However, the conclusions on their distinct oligomeric states are premature. The oligomeric state of caveolins is analysed by density gradients after mild solubilisation with 0,5% Triton X-100. Is it clear that under these mild conditions caveolins are fully solubilized?

The Reviewer raises a very important point, and we share the concerns (especially with the new understanding of caveolin oligomerization and membrane association that comes from the cryoEM structure). From the pioneering description of caveolin oligomers using similar techniques (Monier et al., 1996; Monier et al., 1995; Sargiacomo et al., 1995), several groups have characterized with detail the validity and physiological significance of these gradients; including different detergents, significance of the weights, how tags modify sedimentation, behavior of mutants, temperature blocks, and even aggresome formation (Han et al., 2015; Hayer et al., 2010a; Hayer et al., 2010b; Scheiffele et al., 1998). We are aware that, although very useful comparative and quantitative techniques, these biochemical approaches have intrinsic limitations when raising conclusions.

To address this point, we have now included an additional biochemical criterion based on the combination of non-ionic and ionic detergents (TX and SDS) (Copeland et al., 2017; Hayer et al., 2010a). This biochemical approach complements the previous two and provides additional information on the oligomeric state of caveolin, facilitating the comparative analyses of different caveolin proteins carried out here. We believe that these biochemical assays together with the microscopy studies provide a new understanding of the temporal and spatial process of caveolin maturation and how this is perturbed by specific mutations.

The presence of any lipids or protein bound to caveolins under these conditions will affect their migration and compromise the conclusion regarding their oligomeric state. For example, in Figure 2F it is shown that CAV3 floats to the top of the gradients. Rather than the monomeric/low oligomeric state, this migration pattern probably reflects CAV3 association with the light lipid droplet fraction. In this reviewer's opinion further validation of the assay is required to demonstrate its suitability in determining caveolins oligomeric states, a central point of the manuscript.

We do agree with the Reviewer that these classical biochemical approaches have intrinsic limitations and need cautious interpretation when raising conclusions because these complexes can be affected by lipid binding, presence of other proteins within the oligomers, formation of non-functional aggregates, etc. However, when used in combination with other biochemical and microscopic methods, they are convenient comparative and quantitative tools.

As noticed by Reviewer 2, an important limitation of the 0.5% Triton X-100 sedimentation gradients is that they do not distinguish between “functional oligomers” and “non-functional aggregates”. To determine if caveolins are forming oligomers or non-functional high-molecular-weight aggregates, like those formed by mutants failing to oligomerize (details in the new Figs. 5 and 6), we have applied an additional biochemical criterion. The nature of the complexes formed by *wt* caveolins and mutants has been evaluated after solubilization of cells with a combination of 0.2% TX and 0.4% SDS. Upon these conditions, the 8S oligomers remain as insoluble complexes but aggregates are largely solubilized into monomeric caveolins (Copeland et al., 2017; Hayer et al., 2010a).

Finally, we have previously tested if LD proteins are completely solubilized by these detergents (Bosch et al., 2020; Turro et al., 2006). Thus, at least in the case of CAV3, floatation in these gradients is not due to its association with LDs but to its low oligomeric state. Notice that to float LD proteins, a free detergent and mild homogenization (nitrogen cavitation) are required to preserve protein association with LDs.

3) There are convincing differences in the steady levels of Caveolins in cells treated with proteasome inhibitors or expressing the K*R mutants. However, the connections between oligomerization and ERAD/proteasomal degradation are not obvious. First, the effect of proteasome inhibition in caveolin oligomerization is not directly tested.

We have now included the effect of the proteasomal inhibition mediated by MG132 on the oligomerization/aggregation state of all caveolins and mutants. Proteasomal inhibition, as displayed by non-degradable caveolin mutants, leads to caveolin aggregation and trafficking defects (Figs. 2, 5, and 6).

Second, the results with the K*R mutants is confusing. It is stated that "These mutants retain the capacity to traffic to the PM". However, this is not supported by the data presented in Figure 3, showing that both CAV1 and CAV3 K*R mutations localize to ER and GC.

The K*R mutants have been described by other groups as efficiently transported to the PM (Aslanyan et al., 2023; Hayer et al., 2010b): "*When expressed in CV1 cells, a GFP-tagged form of the mutant readily reached the plasma membrane, where it was detectable as subresolution spots typical for caveolae (Fig. S3 B). Successful caveolae assembly of the mutant was also confirmed by sucrose velocity centrifugation of lysates prepared from cells expressing GFP- or HA-tagged CAV1-K*R (Fig. S3 C).*"

However, the Reviewer is right, and when analyzed by using our "particular" experimental methods these mutants are significantly, but not completely, intracellularly retained, as observed by microscopy, and corroborated biochemically. The most likely explanation for this difference is that those studies used "standard" long-term transfections; in contrast to those studies our three-hour analysis, as illustrated by comparison of CAV1 and CAV3, is demonstrating that the transport of K*R to the PM is much slower than the transport of *wt* CAV1 (explanation in pages 8-9). Our biochemical analysis confirmed that CAV1 K*R forms 8S oligomers but that after three hours, in contrast to CAV1, is largely accumulated as low oligomeric species (Fig. 3).

CAV1 K*R migrates mainly to the bottom of the gradient. It is not clear that this is an oligomer or an aggregate.

The new biochemical criterion demonstrates that these species formed by the K*R mutant are aggregates (Fig. 3).

Third, the lipid regulation experiments could be potentially interesting but no treatment controls should have been included. In addition, control proteins that do not respond to the lipid treatments should have been included in the analysis.

We have included VAP-A, not affected by lipids, and ALDI, a LD resident protein responding like CAV3 (Fig. 3), as mechanistically characterized in lower eukaryotes by (Ruggiano et al., 2016) and now explained in the text (page 9 and 17).

4) The phenotypes of the pathogenic mutations are very subtle mild. It is difficult to appreciate the difference between Fig5C and CAV3 in Fig3D.

We have quantified the colocalization of all caveolins with the relevant organelles (ER, GC, LD, and PM) in confocal microscopy images. Please be aware that these mutants are just expressed for three hours, in contrast to the classical transfections for much longer times and that the effects of the mutants are accumulative. Longer expression times are now included in Fig. S5 to illustrate that aggregation is progressive.

Also the authors make a point that ERAD can clear low oligomeric/monomeric CAV. I would therefore suspect that the pathogenic mutants should be easily cleared since they cannot oligomerize. However, their levels at the top of the velocity gradient seem very abundant and only slightly oligomerize during lengthy expression. It would be important to discuss this discrepancy as to why these mutants are not cleared despite being at a low monomeric conditions. What happens at further time points of expression?

Indeed, all these pathogenic caveolin mutants are characterized by their low protein steady state levels when compared with the *wt* proteins and expressed for longer times (see for example P158 in (Copeland et al., 2017), CAV1-P132L in (Lee et al., 2002), and CAV3-P104L in (Kuga et al., 2011)). The low protein steady state levels point confirm that intracellularly retained caveolins are efficiently degraded. Even at three hours intracellular retained mutants are less stable than *wt* proteins (especially clear for CAV3-P104L), as we observed when optimizing our experimental protocol. We did not mention this point in the previous version, but now additional explanations are added in the text (page 14).

Legend: GFP-tagged caveolins and mutants were expressed for three hours in CTL COS cells or in cells treated with MG132 (low dose first line and high dose second line). Protein steady state levels were determined with anti-GFP antibodies. VAP-A was used as a control of an ER protein exhibiting less proteasomal degradation than caveolins.

Minor comments:

1) Introduction, paragraph 2: "...examine the early proteostasis of caveolin family members and mutants". Proteostasis is jargon with very loose meaning. In this context and at other places of the manuscript I would suggest authors to be more specific about the question they aim to address.

We adopted the term "proteostasis" after reading a recent *Nat Rev Mol Cell Biol* (Hipp et al., 2019) and found a potential link between these processes and the poorly understood and unresolved role of caveolins in long-term pathogenic conditions. We are open to other suggestions, but we believe it does encompass the range of cellular mechanisms involved in regulating caveolin protein levels.

2) Caveolins do not possess a signal peptide or a transmembrane domain. Is it known how they are targeted to the ER? It would be interesting to give additional background information about this.

Original experiments determined that CAV1 protomers do indeed insert into the ER membrane through a "hairpin" hydrophobic domain mediated by the classical translocation machinery. Using an *in vitro* transcription-translation-translocation system, it was determined that insertion into the membrane is co-translational, SRP dependent, and need the docking protein (SRP receptor) and sec61 and TRAM (Monier et al., 1996; Monier et al., 1995) (now explained on page 3).

The new determined 3D structure of the caveolin oligomers suggest that once the rings are formed the association of caveolins with the membranes is much more complex. The ring forms a flat disc

with one extremely hydrophobic face that likely occupies the cytosolic leaflet of the lipid bilayer (Fig. 1 and text).

3) The degradation of Caveolins by ERAD is mentioned multiple times. However, only inhibition of the proteasome, which degrades many substrates, was tested. Therefore, the authors should consider to change their terminology.

Proteasomal degradation instead of ERAD is now used throughout the text.

4) In Figure 3D it would be important to show that CAV3 expressing cells have LDs.

Colocalization of K^{*}R and Plin-3 is included in the new version.

5) In Figure 4 (and others), protein steady state levels are analyzed. This is different from protein stability and the two terms should not be used interchangeably.

“Protein steady state levels” is used instead of “protein stability”.

6) The authors should provide more details on how the expression of XBP1S and CHOP was determined.

More experimental details are included in the new version.

7) It would be interesting to discuss what the roles for CAV3 retention in the ER and on LDs might be in a physiological context.

A new Discussion explaining these, and other conclusions has been written. Unfortunately, the role of caveolins in the ER/LDs encloses an unexpected complexity that we and others have not been able to clarify since our original description in *JCB* 20 year ago (Pol et al., 2001), even though the presence of caveolins on LDs has been repeatedly confirmed by other groups in cell biology, metabolism, and proteomic studies (Parton et al., 2020)(see new Discussion, page 18).

8) Based on the modelling, could the authors speculate how CAV2 would behave if tested in this study.

The Reviewer’s raises a very interesting question. The role of CAV2 in caveola formation is very intriguing and as yet not understood. CAV2 is unable to form functional oligomers and caveolae without CAV1 (Mora et al., 1999; Scheiffele et al., 1998; Sowa et al., 2003). CAV1/CAV2 heterooligomers seem to be involved in formation of basolateral caveolae of epithelia while CAV1 oligomers are involved in apical caveolae formation. We know that CAV2 is also actively degraded by the proteasome (example of IB on the right) and largely accumulates in GC without oligomerizing (Fig. 2). In our view, as predicted by AlphaFold (Figs. 1 and S1) and occurring when expressed in bacteria (Walser et al., 2012), CAV2 should by its own be able to form rings in cells but on the contrary it does not occur. One explanation, which is raised by the results of this manuscript, is that CAV2 is unable to reach the oligomerization sites in the late-GC in the absence of CAV1. This suggestion is actually supported by our study and by previous work showing that CAV2 is present in a monomeric state but becomes oligomeric upon CAV1 expression (Sowa et al., 2003). We now cite these studies and propose this interesting hypothesis in the new Discussion (page 16).

References

- Aslanyan, M.G., C. Doornbos, G.D. Diwan, Z. Anvarian, T. Beyer, K. Junger, S.E.C. van Beersum, R.B. Russell, M. Ueffing, A. Ludwig, K. Boldt, L.B. Pedersen, and R. Roepman. 2023. A targeted multi-proteomics approach generates a blueprint of the ciliary ubiquitinome. *Frontiers in Cell and Developmental Biology*. 11.

- Bosch, M., M. Mari, S.P. Gross, J.C. Fernandez-Checa, and A. Pol. 2011a. Mitochondrial cholesterol: a connection between caveolin, metabolism, and disease. *Traffic*. 12:1483-9.
- Bosch, M., M. Mari, A. Herms, A. Fernandez, A. Fajardo, A. Kassan, A. Giralt, A. Colell, D. Balgoma, E. Barbero, E. Gonzalez-Moreno, N. Matias, F. Tebar, J. Balsinde, M. Camps, C. Enrich, S.P. Gross, C. Garcia-Ruiz, E. Perez-Navarro, J.C. Fernandez-Checa, and A. Pol. 2011b. Caveolin-1 deficiency causes cholesterol-dependent mitochondrial dysfunction and apoptotic susceptibility. *Curr Biol*. 21:681-6.
- Bosch, M., M. Sanchez-Alvarez, A. Fajardo, R. Kapetanovic, B. Steiner, F. Dutra, L. Moreira, J.A. Lopez, R. Campo, M. Mari, F. Morales-Paytuy, O. Tort, A. Gubern, R.M. Templin, J.E.B. Curson, N. Martel, C. Catala, F. Lozano, F. Tebar, C. Enrich, J. Vazquez, M.A. Del Pozo, M.J. Sweet, P.T. Bozza, S.P. Gross, R.G. Parton, and A. Pol. 2020. Mammalian lipid droplets are innate immune hubs integrating cell metabolism and host defense. *Science*. 370.
- Copeland, C.A., B. Han, A. Tiwari, E.D. Austin, J.E. Loyd, J.D. West, and A.K. Kenworthy. 2017. A disease-associated frameshift mutation in caveolin-1 disrupts caveolae formation and function through introduction of a de novo ER retention signal. *Mol Biol Cell*. 28:3095-3111.
- Han, B., A. Tiwari, and A.K. Kenworthy. 2015. Tagging strategies strongly affect the fate of overexpressed caveolin-1. *Traffic*. 16:417-38.
- Hayer, A., M. Stoeber, C. Bissig, and A. Helenius. 2010a. Biogenesis of caveolae: stepwise assembly of large caveolin and cavin complexes. *Traffic*. 11:361-82.
- Hayer, A., M. Stoeber, D. Ritz, S. Engel, H.H. Meyer, and A. Helenius. 2010b. Caveolin-1 is ubiquitinated and targeted to intraluminal vesicles in endolysosomes for degradation. *J Cell Biol*. 191:615-29.
- Hipp, M.S., P. Kasturi, and F.U. Hartl. 2019. The proteostasis network and its decline in ageing. *Nat Rev Mol Cell Biol*. 20:421-435.
- Hulce, J.J., A.B. Coggnetta, M.J. Niphakis, S.E. Tully, and B.F. Cravatt. 2013. Proteome-wide mapping of cholesterol-interacting proteins in mammalian cells. *Nat Methods*. 10:259-64.
- Kim, J.H., J.P. Schleich, Z. Lu, D. Peng, K.C. Reasoner, and C.R. Sanders. 2016. A pH-Mediated Topological Switch within the N-Terminal Domain of Human Caveolin-3. *Biophys J*. 110:2475-2485.
- Kuga, A., Y. Ohsawa, T. Okada, F. Kanda, M. Kanagawa, T. Toda, and Y. Sunada. 2011. Endoplasmic reticulum stress response in P104L mutant caveolin-3 transgenic mice. *Hum Mol Genet*. 20:2975-83.
- Lee, H., D.S. Park, B. Razani, R.G. Russell, R.G. Pestell, and M.P. Lisanti. 2002. Caveolin-1 mutations (P132L and null) and the pathogenesis of breast cancer: caveolin-1 (P132L) behaves in a dominant-negative manner and caveolin-1 (-/-) null mice show mammary epithelial cell hyperplasia. *Am J Pathol*. 161:1357-69.
- Luetterforst, R., E. Stang, N. Zorzi, A. Carozzi, M. Way, and R.G. Parton. 1999. Molecular characterization of caveolin association with the Golgi complex: identification of a cis-Golgi targeting domain in the caveolin molecule. *J Cell Biol*. 145:1443-59.
- Monier, S., D.J. Dietzen, W.R. Hastings, D.M. Lublin, and T.V. Kurzchalia. 1996. Oligomerization of VIP21-caveolin in vitro is stabilized by long chain fatty acylation or cholesterol. *FEBS Lett*. 388:143-9.
- Monier, S., R.G. Parton, F. Vogel, J. Behlke, A. Henske, and T.V. Kurzchalia. 1995. VIP21-caveolin, a membrane protein constituent of the caveolar coat, oligomerizes in vivo and in vitro. *Mol Biol Cell*. 6:911-27.
- Mora, R., V.L. Bonilha, A. Marmorstein, P.E. Scherer, D. Brown, M.P. Lisanti, and E. Rodriguez-Boulan. 1999. Caveolin-2 localizes to the golgi complex but redistributes to plasma membrane, caveolae, and rafts when co-expressed with caveolin-1. *J Biol Chem*. 274:25708-17.
- Park, H.J., J. Jang, K.S. Ryu, J. Lee, S.H. Lee, H.S. Won, E.H. Kim, M.D. Seo, and J.H. Kim. 2021. Structural Interplays in the Flexible N-Terminus and Scaffolding Domain of Human Membrane Protein Caveolin 3. *Membranes (Basel)*. 11.
- Parton, R.G., M.A. Del Pozo, S. Vassilopoulos, I.R. Nabi, S. Le Lay, R. Lundmark, A.K. Kenworthy, A. Camus, C.M. Blouin, W.C. Sessa, and C. Lamaze. 2020. Caveolae: The FAQs. *Traffic*. 21:181-185.
- Pol, A., R. Luetterforst, M. Lindsay, S. Heino, E. Ikonen, and R.G. Parton. 2001. A caveolin dominant negative mutant associates with lipid bodies and induces intracellular cholesterol imbalance. *J Cell Biol*. 152:1057-70.
- Pol, A., F. Morales-Paytuy, M. Bosch, and R.G. Parton. 2020. Non-caveolar caveolins - duties outside the caves. *J Cell Sci*. 133.

- Porta, J.C., B. Han, A. Gulsevin, J.M. Chung, Y. Peskova, S. Connolly, H.S. McHaourab, J. Meiler, E. Karakas, A.K. Kenworthy, and M.D. Ohi. 2022. Molecular architecture of the human caveolin-1 complex. *Sci Adv.* 8:eabn7232.
- Ruggiano, A., G. Mora, L. Buxo, and P. Carvalho. 2016. Spatial control of lipid droplet proteins by the ERAD ubiquitin ligase Doa10. *EMBO J.* 35:1644-55.
- Sargiacomo, M., P.E. Scherer, Z. Tang, E. Kubler, K.S. Song, M.C. Sanders, and M.P. Lisanti. 1995. Oligomeric structure of caveolin: implications for caveolae membrane organization. *Proc Natl Acad Sci U S A.* 92:9407-11.
- Scheiffele, P., P. Verkade, A.M. Fra, H. Virta, K. Simons, and E. Ikonen. 1998. Caveolin-1 and -2 in the exocytic pathway of MDCK cells. *J Cell Biol.* 140:795-806.
- Sowa, G., M. Pypaert, D. Fulton, and W.C. Sessa. 2003. The phosphorylation of caveolin-2 on serines 23 and 36 modulates caveolin-1-dependent caveolae formation. *Proc Natl Acad Sci U S A.* 100:6511-6.
- Turro, S., M. Ingelmo-Torres, J.M. Estanyol, F. Tebar, M.A. Fernandez, C.V. Albor, K. Gaus, T. Grewal, C. Enrich, and A. Pol. 2006. Identification and characterization of associated with lipid droplet protein 1: A novel membrane-associated protein that resides on hepatic lipid droplets. *Traffic.* 7:1254-69.
- Walser, P.J., N. Ariotti, M. Howes, C. Ferguson, R. Webb, D. Schwudke, N. Leneva, K.J. Cho, L. Cooper, J. Rae, M. Floetenmeyer, V.M. Oorschot, U. Skoglund, K. Simons, J.F. Hancock, and R.G. Parton. 2012. Constitutive formation of caveolae in a bacterium. *Cell.* 150:752-63.

May 18, 2023

RE: JCB Manuscript #202204020R-A

Dr. Albert Pol
Consorci Institut D'Investigacions Biomediques August Pi I Sunyer
Lipid Trafficking and Disease Group
Centre de Recerca Biomèdica CELLEX (IDIBAPS), Office B12
Casanova 143, 08036
Barcelona 08036
Spain

Dear Dr. Pol:

Thank you for submitting your revised manuscript entitled "Dynamic early proteostasis of caveolins balances trafficking and degradation to reduce cytotoxicity". Editorially, we appreciate that your revised manuscript addresses many of the original concerns and overall find that the data are of interest for the readership of JCB. However, we agree with some of the reviewer concerns that some conclusions are not fully supported, or have alternative explanations. You might also consider Reviewer 1's comments regarding improving the flow of logic, although this is not an essential point. In your final revision please try to better explain some of the technical questions - notably the difference between cav localization and organelle co-localization metrics; how the quantification of protein abundance was performed to give the reported results, which sometimes seem opposite from the blots; and clarifying controls for the lipid treatment experiments. You must also tone down some conclusions and better discuss caveats and limitations. Pending these changes and final revisions necessary to meet our formatting guidelines (see details below) we would be happy to publish your paper in JCB.

A. MANUSCRIPT ORGANIZATION AND FORMATTING:

- 1) Text limits: Character count for Articles is < 40,000, not including spaces. Count includes abstract, introduction, results, discussion, and acknowledgments. Count does not include title page, figure legends, materials and methods, references, tables, or supplemental legends.
- 2) Figures limits: Articles may have up to 10 main text figures.
- 3) Figure formatting: Scale bars must be present on all microscopy images, * including inset magnifications (you may alternatively indicate the diameter of the inset). * Molecular weight or nucleic acid size markers must be included on all gel electrophoresis. * In order to accommodate readers with red-green color blindness, we ask that you please change any red/green color schemes.
- 4) Statistical analysis: Error bars on graphic representations of numerical data must be clearly described in the figure legend. The number of independent data points (n) represented in a graph must be indicated in the legend. Statistical methods should be explained in full in the materials and methods. For figures presenting pooled data the statistical measure should be defined in the figure legends. Please also be sure to indicate the statistical tests used in each of your experiments (either in the figure legend itself or in a separate methods section) as well as the parameters of the test (for example, if you ran a t-test, please indicate if it was one- or two-sided, etc.). Also, if you used parametric tests, please indicate if the data distribution was tested for normality (and if so, how). If not, you must state something to the effect that "Data distribution was assumed to be normal but this was not formally tested."
- 5) Abstract and title: The abstract should be no longer than 160 words and should communicate the significance of the paper for a general audience. The title should be less than 100 characters including spaces. Make the title concise but accessible to a general readership.
- 6) Materials and methods: Should be comprehensive and not simply reference a previous publication for details on how an experiment was performed. Please provide full descriptions in the text for readers who may not have access to referenced manuscripts.
- 7) Please be sure to provide the sequences for all of your primers/oligos and RNAi constructs in the materials and methods. You

must also indicate in the methods the source, species, and catalog numbers (where appropriate) for all of your antibodies. * Please also indicate the acquisition (e.g. digital imager model) and quantification methods for immunoblotting/western blots.

8) Microscope image acquisition: The following information must be provided about the acquisition and processing of images:

- a. Make and model of microscope
- b. Type, magnification, and numerical aperture of the objective lenses
- c. Temperature
- d. Imaging medium
- e. Fluorochromes
- f. Camera make and model
- g. Acquisition software
- h. Any software used for image processing subsequent to data acquisition. Please include details and types of operations involved (e.g., type of deconvolution, 3D reconstitutions, surface or volume rendering, gamma adjustments, etc.).

10) Supplemental materials: There are strict limits on the allowable amount of supplemental data. Articles may have up to 5 supplemental figures. Please also note that tables, like figures, should be provided as individual, editable files. A summary of all supplemental material should appear at the end of the Materials and methods section.

13) ORCID IDs: ORCID IDs are unique identifiers allowing researchers to create a record of their various scholarly contributions in a single place. At resubmission of your final files, please consider providing an ORCID ID for as many contributing authors as possible.

Please note that JCB now requires authors to submit Source Data used to generate figures containing gels and Western blots with all revised manuscripts. This Source Data consists of fully uncropped and unprocessed images for each gel/blot displayed in the main and supplemental figures. Since your paper includes cropped gel and/or blot images, please be sure to provide one Source Data file for each figure that contains gels and/or blots along with your revised manuscript files. File names for Source Data figures should be alphanumeric without any spaces or special characters (i.e., SourceDataF#, where F# refers to the associated main figure number or SourceDataFS# for those associated with Supplementary figures). The lanes of the gels/blots should be labeled as they are in the associated figure, the place where cropping was applied should be marked (with a box), and molecular weight/size standards should be labeled wherever possible.

Journal of Cell Biology now requires a data availability statement for all research article submissions. These statements will be published in the article directly above the Acknowledgments. The statement should address all data underlying the research presented in the manuscript. Please visit the JCB instructions for authors for guidelines and examples of statements at (<https://rupress.org/jcb/pages/editorial-policies#data-availability-statement>).

B. FINAL FILES:

Thank you for this interesting contribution, we look forward to publishing your paper in Journal of Cell Biology.

Sincerely,

Elizabeth Miller, PhD
Monitoring Editor

Andrea L. Marat, PhD
Senior Scientific Editor

Journal of Cell Biology

Reviewer #1 (Comments to the Authors (Required)):

In the revised manuscript, the authors have performed new experiments and majorly revised the text to respond to the comments raised. These new data and complementary controls have resolved many of the unclarities and followed up on some of the observations to strengthen the conclusions draw in the manuscript. The manuscript now provides a thorough comparison of synthesis and secretory transport of the Caveolin-family of proteins showing that retention signals and oligomerization influence trafficking and degradation in order to avoid ER stress and missfolded protein aggregates. Yet, the presentation of the data could be improved in order to increase readability and clarity. Also, the figures are extensive and some of the supporting data could be moved to supplementary figures to increase readability. Some of the data does not hold up and could be removed as described below.

Comments:

Fig 2 The addition of markers for organelles and additional biochemical analysis has majorly improved the manuscript. If gels of immunoblots are shown here for sedimentation analysis, they could be put in supplementary for following figures when the reader then know what the data is, as a suggestion.

Fig3 and Fig4: address differences between cav1 and cav3 observed in Fig 2, localization and expression levels. I would suggest to change the order of Fig 3 and Fig 4 in order to improve readability. Fig 3 would then specifically address why the trafficking of cav3 is different to cav1 and that this depends on the N-terminus. I would suggest to state that this is a potential retention signal. Are there examples where hydrophobicity close to the COPII binding site affects the strength of the retention signal that could be used to strengthen this? Fig4 would then follow up on the proteasomal degradation and how this affect protein levels. Less protein expression and the need of a retention signal could suggest that Cav3 is more difficult to fold and oligomerize properly.

I might have missed this, but I still don't find how the addition of cholesterol and FFA has been performed and verified. It just says "supplemented with 1.1mM, or 4mM cholesterol"?

Since these data cannot be assessed and the increased levels of cholesterol or FA in ER GC has not been verified, these data should be removed from the manuscript. Also the difference are quite small and these experiments do not influence the overall conclusions for the manuscript.

As a minor comment to Fig4D, I suggest to move the helical prediction of Cav3 to the left of the helical prediction of Cav1 to avoid misunderstanding since the sequence in cav3 is underlined

Analysis of mutants in Fig 5 and Fig 6: The description of these data and conclusions could be described better in the text. While the oligomerization mutants are thoroughly discussed (structural analysis), the P158 frameshift is less described. The frameshift results in a drastic change in 21 aa in the C term. Yet, the authors only focus on the potential KKxx retention signal. This is described as a "Newly generated ER-retrieval signal" Newly feels like the wrong word? Unnatural. Could it be phrased like "the frameshift resulted in an unnatural ER-retrieval signal in the N-terminus? Anyway, the authors interpret that it is this unnatural retrieval signal that causes the effect on trafficking, which then would subsequently affect oligomerization. However, the new reading frame results in a stretch of 21 amino acids being altered. This could also affect structure, lipid binding, oligomerization etc. This could be tested by mutation of the unnatural retrieval signal. Since this has not been addressed, I suggest that the authors should tone down the conclusions.

Also it is not clear what the first section of text under the heading "Impaired proteostasis of newly-synthesized" Refers to and describe?

Reviewer #2 (Comments to the Authors (Required)):

The revised manuscript by Morales-Paytuyi et al. includes several new experiments and significant changes to the text, which now flows well and is clearer. Some of the new data, in particular the co-localization analysis, is also informative. However, despite these improvements, It is unclear if the complex relation between trafficking, degradation and oligomerization of Caveolins has been sufficiently resolved by the study. In this reviewer opinion, the some of the conclusions are not fully supported (for example on the role of lipids in controlling caveolin trafficking) and often based on very small differences. As mentioned above, the inclusion of organelle markers to the ER (calreticulin), GC (TGN46), LD (PLIN3) and PM (Cavin 1) and the co-localization analysis using the Mander's coefficient (in "organelle colocalization") is informative. But how is "Caveolin localization" determined? Does it correspond to the previous quantification? The inclusion of both analysis is confusing, particularly because in some cases "Caveolin localization" and "organelle colocalization" show conflicting results. Without being exhaustive, a few examples are GC localization of CAV3 and the ER localization of CAV2 (Figs 2H and I). In addition, the organelle colocalization analysis using Mander's coefficient show milder differences on the distribution of the different CAV proteins and mutants, and that in some case isn't even significant (see for example Figs 2H vs 2I, 3H vs 3I and 4G vs 4H). Considering these small effects, is it possible that the differences in ER localization results from the differences in expression levels rather than trafficking? This also applies to insolubility of CAVs. Is CAV1 appearing earlier in the insoluble fraction simply because it is expressed to higher levels?

Other unresolved issues are below.

- 1- The effects of lipids in Caveolin treatment remain unconvincing and can have alternative explanations. In addition, it is unclear how the experiment was done and if a control without lipid treatment was included. According to the figure it seems that Control (CTL) correspond to cells non treated with MG132. However, the figure legend refers cells not "supplemented with cholesterol (CHL) or oleic acid (OA)."
- 2- In figure 3 how were protein levels quantified. From 3A it appears clear that MG132 has a much stronger effect on the levels of CAV3 than CAV1. However, quantification in 3B shows the opposite result. How were the quantifications done?
- 3- The earlier concerns regarding the mild phenotypes of the pathogenic mutations remain and only subtle effects in ER stress after expressing Caveolin mutants for 16h.

Other minor points:

- 1- There is a reference to 70S oligomers but it isn't explained what these are. Are they aggregates?
- 2- In Figure 4L, comparison of insolubility should be analysed as in 4C. It is difficult to make conclusions from comparisons between proteins expressed at different levels.

Barcelona, June 3rd, 2023

Re: JCB Manuscript #202204020R-A by Morales-Paytuví et al.

Dear Dr. Marat,

We are delighted that our manuscript was found to be suitable for publication in *The Journal of Cell Biology* subjected to some minor changes. We have now addressed these comments in a revised version as we outline in the point-by-point below. Changes are highlighted in red in the new manuscript. We have also redone the schemes using red and black as colors, to facilitate the reading of the green-red blind audience.

We have slightly modified the title to “*Early proteostasis of caveolins synchronizes trafficking and degradation to prevent toxic aggregation*”. Our feeling is that the word oligomerization should be also included to read “*Early proteostasis of caveolins synchronizes trafficking, degradation, and oligomerization to prevent toxic aggregation*” but in that case it exceeds the allowed limit by 7 characters.

Once again, we would like to thank the editors and referees for their excellent comments and look forward to hearing from you in due course.

Best regards,

Albert Pol and (p.p.) Rob Parton.

Point-by-point response.

Editors' comments.

Editorially, we appreciate that your revised manuscript addresses many of the original concerns and overall find that the data are of interest for the readership of JCB.

However, we agree with some of the reviewer concerns that some conclusions are not fully supported or have alternative explanations.

We sincerely hope that the explanations included in this new version fully support our conclusions, as detailed in this letter.

You might also consider Reviewer 1's comments regarding improving the flow of logic, although this is not an essential point.

We have decided to maintain the gradients in the main figures (we had already 5 supplemental figures) and keep the initial order of figures. It is difficult for us to justify why, when compared with the *wt* forms, the steady state levels of CAV1 N-CAV3 are lower because it is ER-retained or the levels of CAV3 N-CAV1 are higher because it is rapidly exported from the ER without the previous demonstration that caveolins are actively degraded by the proteasome. We think that in this order, Fig. 4 is a convincing confirmation for the data shown in Fig. 3.

In your final revision please try to better explain some of the technical questions - notably the difference between *cav* localization and organelle co-localization metrics;

We recognize that the inclusion of both analyses could be confusing so, we have eliminated the manually annotated quantification. We initially included this “subjective” quantification to avoid intrinsic limitations of colocalization analyses, which are depending on antibodies and thus, affected by unspecific labelling, major affinity for a sub compartment within the organelle, background, etc - problems exacerbated by the low levels of caveolins in these cells and the simultaneous distribution of caveolins between different organelles. In any case, we think that the MOC analysis confirms the major conclusions of the study and agree that it is indeed more “objective”.

how the quantification of protein abundance was performed to give the reported results, which sometimes seem opposite from the blots;

The method to quantify protein steady state levels was explained in detail in the legend of Fig. 2, where the first quantification was shown. These details are now also included in Methods.

and clarifying controls for the lipid treatment experiments.

Although the lipid loading controls could be found in the original papers of our and other groups, following the recommendation of Reviewer#1, we have removed the results related to the lipid-dependent regulation of newly synthesized caveolins.

You must also tone down some conclusions and better discuss caveats and limitations.

These conclusions were based on published data. Some of the suggested experiments were already performed in the original description of the mutants. We apologize for not

explaining these data clearly and now have included additional explanations to support the conclusions.

Pending these changes and final revisions necessary to meet our formatting guidelines (see details below) we would be happy to publish your paper in JCB.

Once again, we are delighted that our manuscript was found to be suitable for publication and truly appreciate the help of editors and reviewers that definitely have helped to improve our work.

Reviewer #1

Fig 2 The addition of markers for organelles and additional biochemical analysis has majorly improved the manuscript. If gels of immunoblots are shown here for sedimentation analysis, they could be put in supplementary for following figures when the reader then know what the data is, as a suggestion.

We also considered this possibility but according to the journal guidelines only 5 supplemental figures are allowed.

Fig3 and Fig4: address differences between cav1 and cav3 observed in Fig 2, localization and expression levels. I would suggest to change the order of Fig 3 and Fig 4 in order to improve readability. Fig 3 would then specifically address why the trafficking of cav3 is different to cav1 and that this depends on the N-terminus. I would suggest to state that this is a potential retention signal. Are there examples where hydrophobicity close to the COPII binding site affects the strength of the retention signal that could be used to strengthen this?

The Reviewer raises an interesting hypothesis. This possibility is now included in the text: “*Whether the CAV3N-terminal alpha helix, which is dispensable for trafficking to the PM or LDs (Luetterforst et al., 1999; Pol et al., 2001), interacts with lipids or proteins in the ER or it is functioning as an ER-retention signal by modulating the affinity of the proximal COPII binding domain (Ma et al., 2017) deserves further studies.*”.

Fig4 would then follow up on the proteasomal degradation and how this affect protein levels.

We thank the reviewer for the suggestion and considered this option carefully. However, it is difficult for us to justify why, when compared with the *wt* forms, the steady state levels of CAV1 N-CAV3 are lower because it is ER-retained or the levels of CAV3 N-CAV1 are higher because it is rapidly exported from the ER without the previous demonstration that caveolins are actively degraded by the proteasome. We think that in this order, Fig. 4 is a convincing confirmation for the data shown in Fig. 3.

Less protein expression and the need of a retention signal could suggest that Cav3 is more difficult to fold and oligomerize properly.

Because our data strongly suggest that oligomerization occurs in the late-GC, a change in folding/oligomerization would not explain the retention of CAV3 in the ER. From our *in silico* analysis (AlphaFold) and from previous studies of CAV1 and CAV3 expressed in bacteria (Walser et al., 2012), we have no indications that differences in oligomerization of CAV3 as compared to CAV1 could explain this effect. Instead, our

results suggest that CAV3 is retained in the ER and less protein reaches the oligomerization sites in the late-GC. This explanation is now included in the text.

I might have missed this, but I still don't find how the addition of cholesterol and FFA has been performed and verified. It just says "supplemented with 1.1mM, or 4mM cholesterol"? Since these data cannot be assessed and the increased levels of cholesterol or FA in ER GC has not been verified, these data should be removed from the manuscript. Also the difference are quite small and these experiments do not influence the overall conclusions for the manuscript.

Following the recommendation of the Reviewer these experiments have been removed from the final version.

As a minor comment to Fig4D, I suggest to move the helical prediction of Cav3 to the left of the helical prediction of Cav1 to avoid misunderstanding since the sequence in cav3 is underlined

The helical prediction of CAV3 was already in the left side of CAV1.

Analysis of mutants in Fig 5 and Fig 6: The description of these data and conclusions could be described better in the text. While the oligomerization mutants are thoroughly discussed (structural analysis),

The PtoL mutants are thoroughly discussed because our new structural analysis demonstrates that rather than disrupting the hairpin topology, as commonly accepted, these mutants affect the formation of caveolin rings (8S oligomers).

the P158 frameshift is less described. The frameshift results in a drastic change in 21 aa in the C term. Yet, the authors only focus on the potential KKxx retention signal. This is described as a "Newly generated ER-retrieval signal" Newly feels like the wrong word? Unnatural. Could it be phrased like "the frameshift resulted in an unnatural ER-retrieval signal in the N-terminus?"

We used "newly generated signal" following the nomenclature used in the title of the original description (Copeland et al., 2017). However, we have included the word "unnatural" in the text to emphasize this concept.

Anyway, the authors interpret that it is this unnatural retrieval signal that causes the effect on trafficking, which then would subsequently affect oligomerization. However, the new reading frame results in a stretch of 21 amino acids being altered. This could also affect structure, lipid binding, oligomerization etc.

The reviewer is correct. However, during the original description of this mutant the authors demonstrated that CAV1-P158 oligomerizes when expressed with wt caveolin suggesting that the new amino acids do not significantly disrupt these processes (Copeland et al., 2017).

This could be tested by mutation of the unnatural retrieval signal. Since this has not been addressed, I suggest that the authors should tone down the conclusions.

The suggested experiment was already done in the original description of the mutant (Copeland et al., 2017). Disruption of the KKYK motif of CAV1-P158 (deletion or KtoA mutation) enables the mutant to traffic to caveolae when expressed in CAV1-KO

cells, proving that the mutant is indeed capable to normally oligomerize. This important detail is now included in the text.

Also it is not clear what the first section of text under the heading "Impaired proteostasis of newly-synthesized" Refers to and describe?

This part of the text has been deleted.

Reviewer #2:

In this reviewer opinion, the some of the conclusions are not fully supported (for example on the role of lipids in controlling caveolin trafficking)

Following the suggestion of Reviewer#1 these experiments have been removed from the final version.

and often based on very small differences.

Although we recognize that some differences may seem small, we want to emphasize again that our analyses have been carried out at extremely short times (3 hours). Although technically challenging, thanks to this new approach, we have been able to demonstrate many previously unknown properties of caveolins and compare the differential affinity of caveolins and mutants for different trafficking pathways, a concept that was not previously considered in the field. Perhaps the most notable example is the significant trafficking differences observed between newly synthesized CAV1 and CAV3, which were previously unnoticed when using the standard 24 hours transfection.

As mentioned above, the inclusion of organelle markers to the ER (calreticulin), GC (TGN46), LD (PLIN3) and PM (Cavin 1) and the co-localization analysis using the Mander's coefficient (in "organelle colocalization") is informative. But how is "Caveolin localization" determined? Does it correspond to the previous quantification? The inclusion of both analysis is confusing, particularly because in some cases "Caveolin localization" and "organelle colocalization" show conflicting results.

We recognize that the inclusion of both analyses could be confusing, so we have eliminated the manually annotated quantification.

Without being exhaustive, a few examples are GC localization of CAV3 and the ER localization of CAV2 (Figs 2H and I). In addition, the organelle colocalization analysis using Mander's coefficient show milder differences on the distribution of the different CAV proteins and mutants, and that in some case isn't even significant (see for example Figs 2H vs 2I, 3H vs 3I and 4G vs 4H).

We initially included this "subjective" quantification to avoid intrinsic limitations of colocalization analyses, which are depending on antibodies and thus, affected by unspecific labelling, major affinity for a sub compartment within the organelle, background, etc - problems exacerbated by the low levels of caveolins in these cells and the simultaneous distribution of caveolins between different organelles. In any case, we think that the MOC analysis confirms the major conclusions of the study and agree that it is indeed more "objective".

Considering these small effects, is it possible that the differences in ER localization results from the differences in expression levels rather than trafficking? This also

applies to insolubility of CAVs. Is CAV1 appearing earlier in the insoluble fraction simply because it is expressed to higher levels?

To avoid the concern raised now by the reviewer, all the proteins used in this work were cloned in an identical pEGFP plasmid and, in principle, should be identically expressed. As an internal control, in the initial experiments, we quantified the levels of transfected GFP-proteins by qPCR with primers recognizing GFP. We did not see any difference in the RNA levels of the different caveolins (see graph) suggesting that the three proteins are identically expressed.

Further, an identical protein steady levels were observed when caveolins were expressed within a MycFlag plasmid to demonstrate that stability/trafficking is not dependent on the tag. To quantify the steady state levels of caveolins, GFP-tagged caveolins were transfected in parallel, to avoid variability between experiments. To rule out that the cell type is differentially affecting expression/stability, GFP- and mycflag-tagged caveolins were additionally transfected in a model muscle cell line with identical results.

In conclusion, we believe that we have proved that the differential steady state levels and trafficking of newly synthesized caveolins is due to the proteostatic mechanisms demonstrated here rather than to different expression levels.

Other unresolved issues are below.

1- The effects of lipids in Caveolin treatment remain unconvincing and can have alternative explanations. In addition, it is unclear how the experiment was done and if a control without lipid treatment was included. According to the figure it seems that Control (CTL) correspond to cells non treated with MG132. However, the figure legend refers cells not "supplemented with cholesterol (CHL) or oleic acid (OA)."

Following the suggestion of Reviewer#1 this experiment has been removed from the final version.

2- In figure 3 how were protein levels quantified. From 3A it appears clear that MG132 has a much stronger effect on the levels of CAV3 than CAV1. However, quantification in 3B shows the opposite result. How were the quantifications done?

We apologize for showing an IB that was not representative and thank the Reviewer for noticing this. We have prepared and included a new IB that better reflects the quantification (average of 8 independent experiments). The method to quantify protein

steady state levels was explained in detail in the legend of Fig. 2 when the first quantification was shown. It is now additionally explained in the Methods section.

3- The earlier concerns regarding the mild phenotypes of the pathogenic mutations remain and only subtle effects in ER stress after expressing Caveolin mutants for 16h.

As in trafficking and stability analyses, in these experiments we prioritized the shorter time when the stress was significantly observable. That intracellularly retained caveolin mutants cause ER stress has been corroborated in mice studies (Kuga et al., 2011). The ER stress caused by mutants increases over the time as does the formation of protein aggregates. We have slightly reworded that sentence to make this point clearer.

Other minor points:

1- There is a reference to 70S oligomers but it isn't explained what these are. Are they aggregates?

The 70S oligomers were fully explained in the Introduction (second paragraph; “Current schemes suggest that the 8S complexes would be rapidly concentrated in ER-exit sites to be transported by COPII vesicles into the GC and assemble 70S oligomers (~150 protomers) that accrue an estimated 20,000 cholesterol molecules (Hayer et al., 2010; Ortegren et al., 2004).”). The 70S oligomers are not aggregates but the current view is that they are higher order oligomers containing approximately 150 caveolin protomers, enriched in cholesterol, formed in the late GC, and rapidly transported to the PM. An additional sentence regarding the possibility of the existence of other high ordered caveolins has been now included in the Introduction.

2- In Figure 4L, comparison of insolubility should be analysed as in 4C. It is difficult to make conclusions from comparisons between proteins expressed at different levels.

The analysis has been included.

- Copeland, C.A., B. Han, A. Tiwari, E.D. Austin, J.E. Loyd, J.D. West, and A.K. Kenworthy. 2017. A disease-associated frameshift mutation in caveolin-1 disrupts caveolae formation and function through introduction of a de novo ER retention signal. *Mol Biol Cell*. 28:3095-3111.
- Hayer, A., M. Stoeber, C. Bissig, and A. Helenius. 2010. Biogenesis of caveolae: stepwise assembly of large caveolin and cavin complexes. *Traffic*. 11:361-82.
- Kuga, A., Y. Ohsawa, T. Okada, F. Kanda, M. Kanagawa, T. Toda, and Y. Sunada. 2011. Endoplasmic reticulum stress response in P104L mutant caveolin-3 transgenic mice. *Hum Mol Genet*. 20:2975-83.
- Luetterforst, R., E. Stang, N. Zorzi, A. Carozzi, M. Way, and R.G. Parton. 1999. Molecular characterization of caveolin association with the Golgi complex: identification of a cis-Golgi targeting domain in the caveolin molecule. *J Cell Biol*. 145:1443-59.
- Ma, W., E. Goldberg, and J. Goldberg. 2017. ER retention is imposed by COPII protein sorting and attenuated by 4-phenylbutyrate. *Elife*. 6.
- Ortegren, U., M. Karlsson, N. Blazic, M. Blomqvist, F.H. Nystrom, J. Gustavsson, P. Fredman, and P. Stralfors. 2004. Lipids and glycosphingolipids in caveolae and

- surrounding plasma membrane of primary rat adipocytes. *Eur J Biochem.* 271:2028-36.
- Pol, A., R. Luetterforst, M. Lindsay, S. Heino, E. Ikonen, and R.G. Parton. 2001. A caveolin dominant negative mutant associates with lipid bodies and induces intracellular cholesterol imbalance. *J Cell Biol.* 152:1057-70.
- Walser, P.J., N. Ariotti, M. Howes, C. Ferguson, R. Webb, D. Schwudke, N. Leneva, K.J. Cho, L. Cooper, J. Rae, M. Floetenmeyer, V.M. Oorschot, U. Skoglund, K. Simons, J.F. Hancock, and R.G. Parton. 2012. Constitutive formation of caveolae in a bacterium. *Cell.* 150:752-63.